# Spatial specificity of auxin responses coordinates wood formation

Klaus Brackmann[1], Jiyan Qi[2], Michael Gebert[2], Virginie Jouannet[2], Theresa Schlamp[2], Karin Grünwald[1], Eva-Sophie Wallner[2], Daria D. Novikova[3,4,5], Victor G. Levitsky[3,4], Javier Agusti[1,6], Pablo Sanchez[1], Jan U. Lohmann[2] & Thomas Greb [2]

Spatial organization of signalling events of the phytohormone auxin is fundamental for maintaining a dynamic transition from plant stem cells to differentiated descendants. The cambium, the stem cell niche mediating wood formation, fundamentally depends on auxin signalling but its exact role and spatial organization is obscure. Here we show that, while auxin signalling levels increase in differentiating cambium descendants, a moderate level of signalling in cambial stem cells is essential for cambium activity. We identify the auxin-dependent transcription factor ARF5/MONOPTEROS to cell-autonomously restrict the number of stem cells by directly attenuating the activity of the stem cell-promoting *WOX4* gene. In contrast, ARF3 and ARF4 function as cambium activators in a redundant fashion from outside of *WOX4*-expressing cells. Our results reveal an influence of auxin signalling on distinct cambium features by specific signalling components and allow the conceptual integration of plant stem cell systems with distinct anatomies.

---

[1] Gregor Mendel Institute (GMI), Austrian Academy of Sciences, Vienna Biocenter (VBC), Dr. Bohr-Gasse 3, 1030 Vienna Austria. [2] Centre for Organismal Studies (COS), Heidelberg University, Im Neuenheimer Feld 230, 69120 Heidelberg Germany. [3] Novosibirsk State University, 2 Pirogova Street, Novosibirsk 630090, Russian Federation. [4] Institute of Cytology and Genetics, 10 Lavrentyeva Avenue, Novosibirsk 630090, Russian Federation. [5] Department of Agrotechnology and Food Sciences, Subdivision Biochemistry, Wageningen University and Research Center, Stippeneng 4, 6708WE Wageningen, The Netherlands. [6] Present address: Instituto de Biología Molecular y Celular de Plantas (IBMCP-CSIC), Universidad Politecnica de Valencia, Ingeniero Fausto Elio s/n, 46022 Valencia, Spain. Correspondence and requests for materials should be addressed to T.G. (email: thomas.greb@cos.uni-heidelberg.de)

In multicellular organisms, communication between cells is essential for coordinated growth and determination of cell fate. In plants in particular, the flexible regulation of cellular properties by hormone signalling is important throughout the whole life cycle. This is because plants are sessile and continuously adapt their growth and development to their local environment. The basis of this plastic growth mode are local stem cell niches at the tips and along plant growth axes, called meristems[1]. The tip-localized shoot and root apical meristems (SAM and RAM, respectively) are essential for primary, or longitudinal, growth of shoots and roots, respectively. In turn, the vascular cambium is the predominant lateral meristem forming a cylinder of indeterminate stem cells at the periphery of growth axes and mediating radial growth by producing the vascular tissues phloem and xylem in a bidirectional manner[2,3]. This production is the basis of wood formation and is thus essential for the accumulation of a large proportion of terrestrial biomass.

The plant hormone auxin plays pivotal roles in local patterning and maintenance of stem cell niches in the SAM and RAM. In the SAM, auxin signalling is low in stem cells and increases during recruitment of cells for organ formation[4–6]. Cell wall modulation and the formation of vascular strands are two aspects promoted by auxin in this context[7,8]. In contrast, a maximum of auxin signalling is present in the quiescent centre, and the surrounding stem cells in the RAM and cell differentiation is, at least partly, driven by a decrease in signalling levels[9,10]. Therefore, the functions of auxin in both meristems are different and adapted to distinct niche requirements.

For the cambium, the role of differential auxin signalling along the radial sequence of tissues is still obscure. In *Arabidopsis* stems, apex-derived auxin is transported basipetally and distributed laterally across the cambial zone by the auxin exporters PIN-FORMED1 (PIN1), PIN3, PIN4 and PIN7[11,12]. Indeed, direct auxin measurements in *Populus* and *Pinus* trees showed that the concentration of the major endogenous auxin indole-3-acetic acid (IAA) peaks in the centre of the cambial zone and gradually declines towards differentiating xylem and phloem cells[13–15]. This observation prompted the idea that, in analogy to the situation in the RAM, radial auxin concentration gradients contribute to the transition of cambium stem cells to secondary vascular tissues[16,17]. Consistently, ubiquitous repression of auxin responses by expressing a stabilized version of the auxin response inhibitor PttIAA3 reduces the number of cell divisions in the cambium region of hybrid aspen trees[18]. In addition, however, the zone of anticlinal cell divisions characteristic for cambial stem cells is enlarged in PttIAA3-overexpressing trees. This suggests that auxin signalling not only promotes cambium proliferation but also spatially restricts stem cell characteristics within the cambium area[12,18,19]. Indeed, especially xylem formation is associated with a local increase of auxin signalling in other contexts[10,20–22] that supports a role of auxin in the recruitment of cells for differentiation similarly as in the SAM. Therefore, it is currently unclear whether auxin signalling is predominantly associated with stem cell-like features or cell differentiation in the context of radial plant growth or how a positive effect on cambium proliferation and on the differentiation of vascular tissues is coordinated.

As a central cambium regulator, the WUSCHEL-RELATED HOMEOBOX4 (WOX4) transcription factor imparts auxin responsiveness to the cambium[23]. Equivalent to the role of WUSCHEL (WUS) and WOX5 in the SAM and RAM[24,25], respectively, WOX4 activity maintains stem cell fate[23,26]. In turn, *WOX4* transcription is stimulated by the leucine-rich repeat receptor-like kinase (LRR-RLK) PHLOEM INTERCALATED WITH XYLEM (PXY). Importantly, the expression domains of the *WOX4* and *PXY* genes presumably overlap and are considered to mark cambium stem cells[23,26–28]. However, a bipartite organization of the cambium zone was shown recently with *PXY* being expressed only in the proximal (xylem-facing) part[29]. Whether this organization reflects the existence of two distinct stem cell pools feeding xylem and phloem production, respectively, has still to be determined.

Here we identify functional domains of auxin signalling in the *Arabidopsis* cambium by local short-term modulation of auxin biosynthesis and signalling. We reveal that, while cambial stem cells do not appear to be a site of elevated auxin signalling, auxin signalling in these cells is required for cambium activity. By analysing transcriptional reporters and mutants of vasculature-associated AUXIN RESPONSE FACTORs (ARFs), we identify ARF3, ARF4 and ARF5 as cambium regulators with different tissue-specificities as well as distinct roles in cambium regulation. Remarkably, whereas ARF3 and ARF4 act redundantly as more general cambium promoters, ARF5 acts specifically in cambium stem cells. In-depth analysis of the auxin- and ARF5-dependent transcriptome in those cells, together with protein–DNA binding assays and genetic analyses, demonstrates that the ARF5-dependent attenuation of *WOX4* is an essential aspect of auxin signalling during cambium regulation.

## Results

**Auxin responses in stem cells stimulate cambium activity.** In *Arabidopsis* stems, the activity of the common auxin response marker *pDR5rev:GFP*[30] was detected in vascular tissues (phloem and xylem) and cortical cells prior and during cambium initiation (Fig. 1a, Supplementary Fig. 1a)[23]. However, there was no overlap with *pWOX4:YFP*[23] or *pPXY:CFP*[28] reporter activities, the two canonical markers for cambium stem cells (Fig. 1a–c, Supplementary Fig. 1a-c)[23]. This suggested that auxin signalling in stem cells occurs at low levels or is even absent. To decide between both possibilities, we generated a plant line expressing an endoplasmatic reticulum (ER)-targeted Yellow Fluorescent Protein (YFP) under the control of the high-affinity *DR5revV2* promoter, which recapitulated the pattern of *DR5revV2* activity previously reported in roots (Supplementary Fig. 1d-f)[31]. In the second internode of elongated shoots, *pDR5rev:GFP* and *pDR5revV2:YFP* activities were congruent but *pDR5revV2:YFP* activity also included the whole cortex as well as cambium cells marked by *pPXY:CFP* activity (Supplementary Fig. 1g-i). Immediately above the uppermost rosette leaf (denoted as stem base throughout the text), stem anatomy shows a secondary configuration, which is characterized by a continuous domain of cambium activity[23]. At this position, the expression domain of *pDR5revV2:YFP* was again broader than the domain of *pDR5rev:GFP* activity substantially overlapping with *pPXY:CFP* activity (Fig. 1d–f). Based on these observations, we concluded that the auxin signalling machinery is active in *PXY*-positive cambial stem cells.

To see whether auxin levels in *PXY*-positive cells were positively correlated with cambium activity, we used the *WOX4* promoter, whose activity fully recapitulated the *PXY* promoter activity in the cambium (Supplementary Fig. 2a-i), for expressing a bacterial tryptophan monooxygenase (iaaM) in an inducible manner[32]. iaaM converts endogenous tryptophan to the IAA precursor indole-3-acetamide and was used before to boost endogenous IAA levels in *Arabidopsis*[33]. As a read out for cambium activity, we determined the amount of interfascicular cambium-derived (ICD) tissues (Supplementary Fig. 3a, b)[34]. Indeed, ethanol-based iaaM induction substantially stimulated the production of ICD tissues (Fig. 1g, h, m, Supplementary Fig. 3c-g), demonstrating that an increase of auxin biosynthesis in *PXY*-positive stem cells stimulates cambium activity.

To determine to which extent downstream components of the auxin signalling cascade are required in those cells, we blocked ARF activity by expressing a dexamethasone (Dex)-inducible variant of the stabilized AUX/IAA protein BODENLOS (Myc-GR-bdl)[35] under the control of the PXY promoter. Consistent with a role of ARF activity in cambium regulation, Dex

treatments of pPXY:Myc-GR-bdl plants resulted in a strongly reduced amount of ICD tissues at the stem base (Fig. 1i, j, m) but not in an altered overall growth habit (Supplementary Fig. 3h). Strikingly, Dex-treated pPXY:Myc-GR-bdl plants showed an even more pronounced repression of IC activity than the inhibition using the BDL promoter[35] (Fig. 1k, l, m, Supplementary Fig. 3i) whose activity was very broad including also PXY-positive cells (Supplementary Fig. 4a-f). These observations indicated that local auxin signalling in PXY-positive stem cells stimulates cambium activity.

**ARF genes are expressed in cambium-associated cells.** To identify ARFs active in cambium stem cells, we mined public transcriptome datasets and found the ARF3/ETTIN, ARF4 and ARF5/MONOPTEROS genes to be co-induced with WOX4 and PXY during cambium initiation[23]. Indeed, pARF3:YFP, pARF4:YFP and pARF5:YFP promoter reporters were active in cambium-related cells at the stem base and the second internode (Fig. 2a, d, g, Supplementary Fig. 5a, d, g). However, while pARF3:YFP and pARF4:YFP reporters were active in rather broad domains including the phloem, the xylem and, partly, pPXY:CFP-positive cells (Fig. 2a-f, Supplementary Fig. 5a-f), pARF5:YFP was exclusively active in cells marked by pPXY:CFP activity (Fig. 2g-i, Supplementary Fig. 5g-i). Moreover, in second internodes, pARF3:YFP and pARF4:YFP activities were both detected in the starch sheath, the innermost cortical cell layer that is considered to serve as the origin of the IC (Supplementary Fig. 5c, f arrows)[37,38], while pARF5:YFP activity was restricted to vascular bundles (Supplementary Fig. 5i). Indicating also a temporal difference between ARF3/4 and ARF5 activities, pARF3:YFP and pARF4:YFP reporters were active together with pPXY:CFP in interfascicular regions at positions approximately 5 mm above the stem base (Supplementary Fig. 6a-f) where cortical cells start dividing to form the IC[34]. In contrast, no pARF5:YFP activity was detected in the same cortical cells (Fig. 2j-l). This observation suggested that ARF5 expression follows the expression of PXY during cambium initiation and is not active during early steps of cambium initiation. Consistently, in pxy-4 mutants where IC formation is largely absent in stems (see below[36]) a pARF5:mCherry promoter reporter was only active in vascular bundles but not in interfascicular regions (Supplementary Fig. 6g-i). Taken together, these observations were in line with a role of ARF3 and ARF4 as

**Fig. 1** : Auxin signalling is required in cambium stem cells. **a–f** Confocal analyses of stem bases from plants containing the auxin response markers pDR5rev:GFP (**a–c**) or pDR5revV2:YFP (**d–f**) and the stem cell marker pPXY:CFP. Overlapping foci between pPXY:CFP (red) and the respective auxin response marker activities are marked by arrows (**c**, **f**). Asterisks mark the vascular bundles. Size bars represent 100 μm. Propidium iodide (PI) staining in blue. **g-l** Toluidine blue stained cross sections at the stem base of wild type (**g**) pWOX4:AlcR; pAlcA:iaaM (**h**), pPXY:Myc-GR-bdl (**i**, **j**) and pBDL:Myc-GR-bdl (**k**, **l**) plants after long-term EtOH (**g**, **h**), mock (**i**, **k**) or Dex (**j**, **l**) treatment. Interfascicular regions are shown and interfascicular cambium-derived tissues (ICD) are marked (red bar). Size bars represent 50 μm. **m** Quantification of ICD tissue extension at the stem base of wild type, pWOX4:AlcR; pAlcA:iaaM (WOX4»iaaM), pPXY:Myc-GR-bdl and pBDL: Myc-GR-bdl plants after long-term EtOH (wild type and WOX4»iaaM), mock (grey) or Dex (yellow) treatment. Student's T-test (pPXY:Myc-GR-bdl (line 2) $p = 9.82E-06$ and pBDL:Myc-GR-bdl $p = 0.001$ and $p = 0.03$) or Welch's T-test (wild type and WOX4»iaaM $p = 9.24E-09$ and pPXY:Myc-GR-bdl (line 1) $p = 5.2E-06$) were performed comparing wild type and WOX4»iaaM and mock and Dex, respectively (Sample sizes $n = 8$–16). The T-bars that extend from the boxes (whiskers) are expected to include 95% of the data. Significance is indicated by asterisks

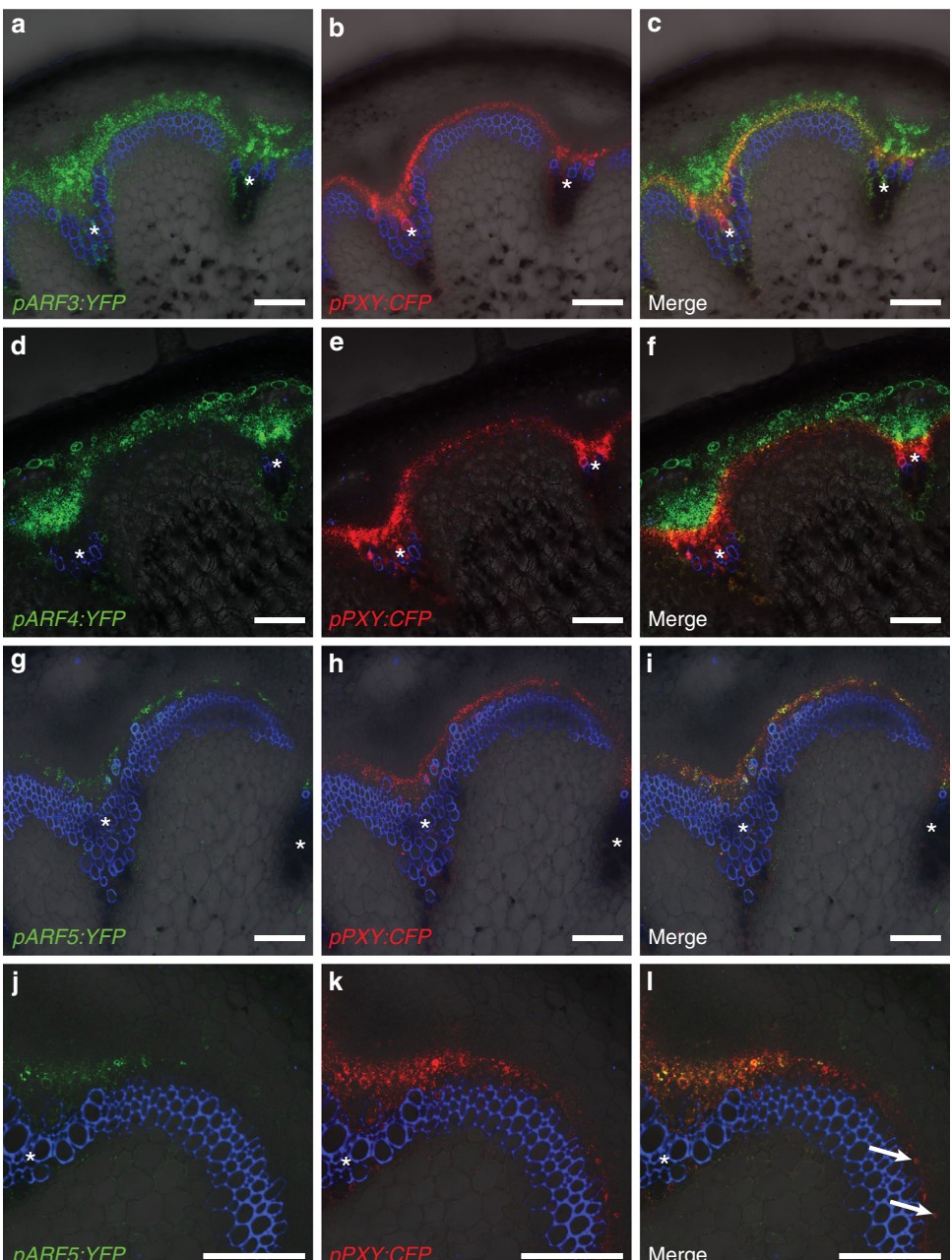

**Fig. 2** *ARF* genes show distinct expression patterns in the cambium area. **a–i** Confocal analyses of stem bases of plants containing *pARF3:YFP* (**a–c**), *pARF4: YFP* (**d–f**) and *pARF5:YFP* (**g–i**), respectively, and the stem cell marker *pPXY:CFP*. Asterisks mark the vascular bundles. Size bars represent 100 μm. **j–l** Confocal analyses of cross-sections from 5 mm above the stem base (transition zone) of plants containing *pARF5:YFP* and the stem cell marker *pPXY:CFP*. Arrows mark cells in the interfascicular region displaying CFP (red) but no YFP (green) activity. Size bars represent 100 μm. PI staining in blue

promoters of cambium activity and a role of *ARF5* as a modulator of the established cambium.

**ARF control of cambium proliferation.** To find indications for these roles, we analysed cambium activity in mutants for the respective *ARF* genes (Supplementary Fig. 7a-e). Consistent with a positive effect of *ARF3*, *arf3* mutants[37,39] showed significantly reduced cambium activity (Fig. 3a, b, e, f, m). This reduction was further increased upon depletion of *ARF4* activity by introducing the *arf4-2* mutation[37] into the respective *arf3* mutant backgrounds (Fig. 3c, d, g, h, m) although the primary stem conformation was similar as in wild type (Supplementary Fig. 7f-m). Consequently, we concluded that cambium activity is positively

regulated by *ARF3* and *ARF4*, which, as in other contexts[37,38], act in a concerted fashion.

In contrast, cambium activity was enhanced in the hypomorphic *arf5* mutant *mp-S319*[40] (Fig. 3i, j, n, Supplementary Fig. 7a, c, n, o), suggesting that *ARF5* counteracts cambium proliferation. Enhanced cambium activity was also found in *mp-S319* hypocotyls (Supplementary Fig. 7r-t) arguing against the possibility that increased cambium activity was observed in stems due to an earlier cambium initiation in interfascicular regions. To confirm a negative impact of *ARF5* on cambium activity, we generated adult plants of the strong *ARF5* loss-of-function mutant *mp-B4149*, which is usually seedling lethal[41], and wild-type plants through tissue culture[42,43] (Supplementary Fig. 7d, e, p, q). Indeed, *mp-B4149* plants showed enhanced ICD formation

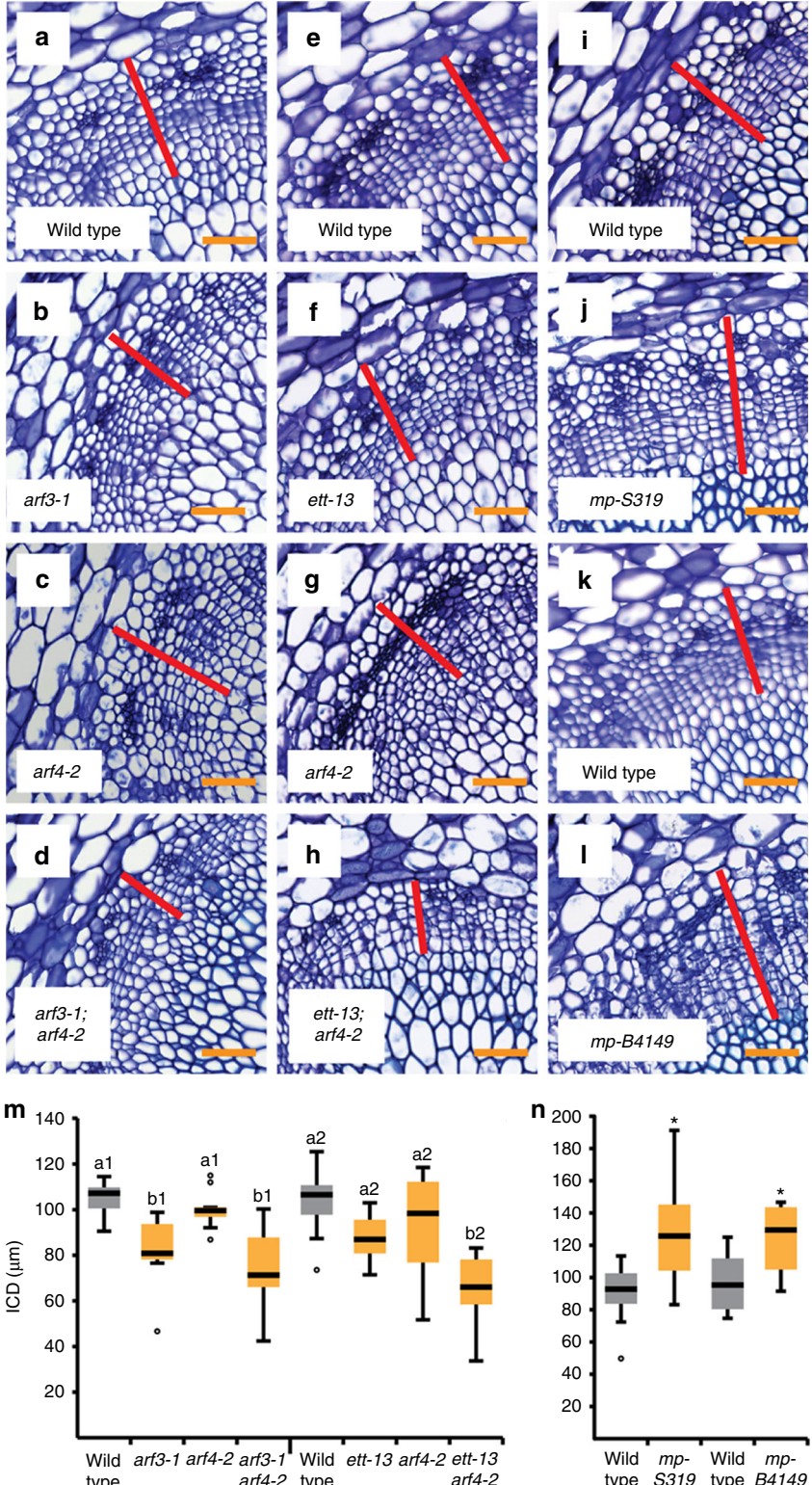

**Fig. 3** *ARF* genes fulfil opposite functions in cambium regulation. **a**–**l** Toluidine blue-stained cross-sections at the stem base of wild-type and *arf* single and multiple mutant plants. Genotypes are indicated. Interfascicular regions are shown and interfascicular cambium-derived tissues (ICD) are marked (red bar). Size bars represent 50 μm. **m**, **n** Quantification of ICD tissues at the stem base of wild-type, *arf3/4* single and double mutants (**m**) and *arf5/mp* mutant plants (**n**). Statistical groups indicated by letters were determined by one-way ANOVA with post hoc Bonferroni (CI 95%; Sample size $n = 8$–10) (**m**). Student's *T*-test was performed comparing wild type and *mp-S319* ($p = 0.003$) and wild type and *mp-B4149* ($p = 0.03$), respectively (Sample sizes $n = 4$–12) (**n**). The T-bars that extend from the boxes (whiskers) are expected to include 95% of the data. Significance is indicated by asterisks

comparable to *mp-S319* mutants (Fig. 3k, l, n). Further confirming a negative effect of ARF5 on cambium activity, ubiquitous expression of a Dex-dependent GR-ARF5 protein fusion using the 35S promoter[44] led to significantly reduced tissue production under long-term induction (Fig. 4a–c, j).

To test whether the identified ARFs function in *PXY*-positive stem cells, we first employed the *PXY* promoter to express GR-ARF5ΔIII/IV, a truncated variant of ARF5 lacking the domains III and IV releasing it from AUX/IAA-based repression[45]. Indeed, long-term Dex treatment of *pPXY:GR-ARF5ΔIII/IV* plants resulted in reduced cambium proliferation (Fig. 4d–f, j) indicating a stem cell-specific role of ARF5. In contrast, the same treatment of a *pPXY:GR-ARF3* line did not influence cambium activity

(Fig. 4g–i, j, see below) arguing against a rate-limiting role of the non-AUX/IAA-dependent[46] ARF3 protein in those cells. Collectively, we concluded that ARF3 and ARF4 on one side and ARF5 on the other side represent two subgroups of ARF transcription factors with differences in both their spatio-temporal expression and roles in cambium regulation.

**ARF5 restricts the number of undifferentiated cambium cells.** To dissect the ARF5-dependent control of cambium stem cells, we took advantage of the DEX-inducibility of our *pPXY:GR-ARF5ΔIII/IV* and of a *p35S:Myc-GR-bdl* line. By determining transcript abundance of the direct ARF5 targets *ATHB8* and *PIN1*[47,48] at different time points after Dex treatment, 3 h of treatment was identified as being optimal for observing short-term effects on gene activity (Supplementary Fig. 8a, b). After determining genome-wide transcript profiles at that time point, we identified a common group of 600 genes with altered transcript levels in both the *pPXY:GR-ARF5ΔIII/IV* and the *p35S:Myc-GR-bdl* line ($p < 0.01$ according to Fisher's Exact Test; Fig. 4k and Supplementary Data 1). The 600 genes represented various functional categories, including primary auxin response (IAAs, SAURs, GH3s), xylem and phloem formation (IAA20 and IAA30[22], REV[12], CVP2 and CVL1[49]) and cell-wall modifications (PMEs and EXPs) (Supplementary Fig. 8c, d; Supplementary Data 2). Moreover, the 312 genes that were induced by *pPXY:GR-ARF5ΔIII/IV* and repressed by *p35S:Myc-GR-bdl* (Fig. 4k; Supplementary Data 1) overlapped significantly with a previously published set of ARF5-inducible genes from seedlings[48] (Supplementary Fig. 8e), indicating that we indeed revealed ARF5-dependent genes in stems. Strikingly, while our expectation was that genes, which are induced by GR-ARF5ΔIII/IV induction, would be repressed by the auxin signalling repressor *bdl* and vice versa, we observed 144 genes (24%) that were either induced (73 genes) or repressed (71 genes) by both transgenes (Fig. 4k). This indicated that in *PXY*-positive cells ARF5 antagonizes the effect of overall auxin signalling on a substantial subset of target genes. Since we observed opposing effects of ARF5 and overall canonical auxin signalling on cambium activity, we suspected that genes integrating these effects are among the 144 genes behaving in an unexpected manner. Interestingly, 11 genes out of the 144 were

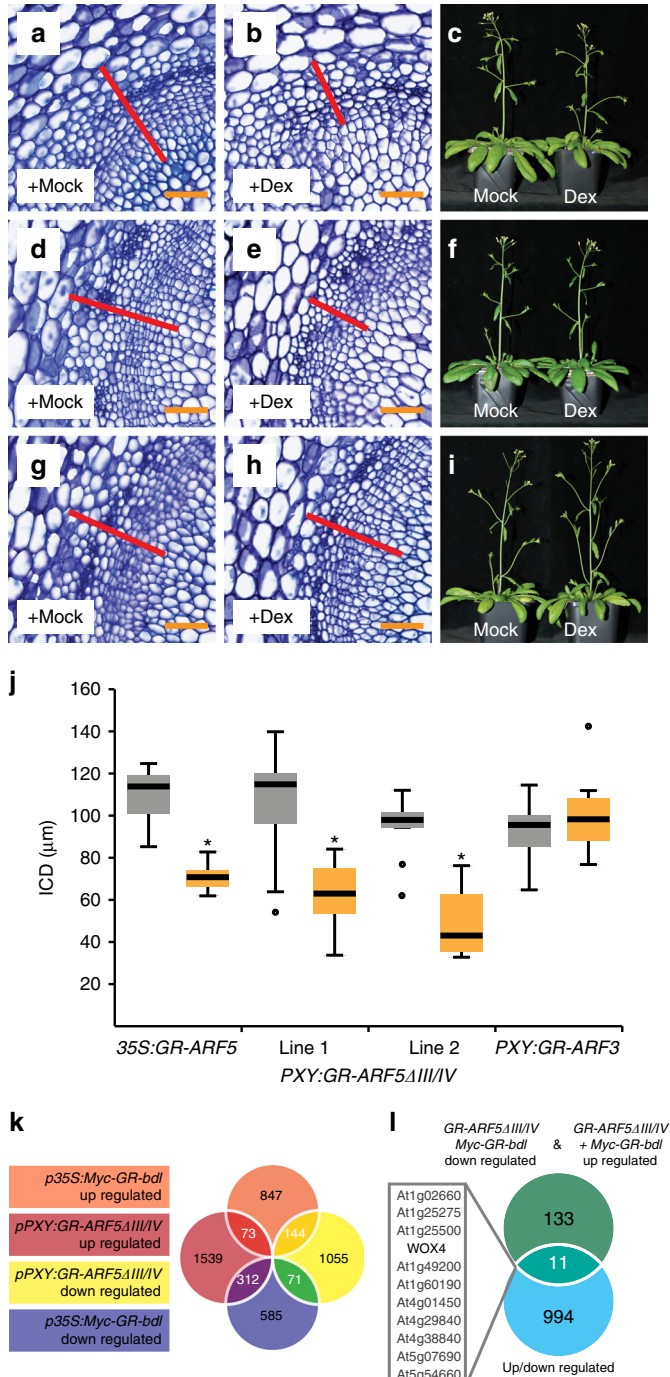

**Fig. 4** ARF5 attenuates cambium proliferation. **a**, **b**, **d**, **e**, **g**, **h** Toluidine blue-stained cross-sections at the stem base of *p35S:GR-ARF5* (**a**, **b**), *pPXY:GR-ARF5ΔIII/IV* (**d**, **e**) and *pPXY:GR-ARF3* (**g**, **h**) plants after long-term mock (**a**, **d**, **g**) or Dex (**b**, **e**, **h**) treatment. Interfascicular regions are shown and ICD tissues are marked (red bar). Size bars represent 50 μm. **c**, **f**, **i** Overview pictures of *p35S:GR-ARF5* (**c**), *pPXY:GR-ARF5ΔIII/IV* (**f**) and *pPXY:GR-ARF3* (**i**) plants after long-term mock or Dex treatment. **j** Quantification of ICD tissue extension at the stem base of *p35S:GR-ARF5*, *pPXY:GR-ARF5ΔIII/IV* and *pPXY:GR-ARF3* plants after long-term mock (grey) or Dex (yellow) treatment. Student's *T*-test (*p35S:GR-ARF5* $p = 3.03E-05$, *pPXY:GR-ARF5ΔIII/IV* (line 2) $p = 1.34E-05$, *pPXY:GR-ARF3* $p = 0.37$) and Welch's *T*-test (*pPXY:GR-ARF5ΔIII/IV* (line 1) $p = 9.72E-04$) were performed comparing mock and Dex treatment (Sample sizes $n = 6–10$). The *T*-bars that extend from the boxes (whiskers) are expected to include 95% of the data. Significance is indicated by the asterisk. **k** Venn diagram of RNA sequencing results from RNA obtained from second internodes of *p35S:Myc-GR-bdl* and *pPXY-GR-ARF5ΔIII/IV* plants after mock or Dex treatment. Identifiers of 600 overlapping genes are shown in Supplementary Data 1. **l** Comparison of the 144 unexpectedly acting genes and the group of genes upregulated or downregulated during cambium initiation[36]. Non-random degree of the overlap was tested by using VirtualPlant 1.3 GeneSect with a cut-off *p*-value < 0.05 and the *Arabidopsis thaliana* Columbia (TAIR10) genome as background population

also found in a set of genes that are differentially expressed during IC formation[36], one of them being *WOX4* that was repressed by both GR-ARF5ΔIII/IV and Myc-GR-bdl induction (Fig. 4l).

To investigate whether the observed effects of altered auxin signalling levels on gene transcription were based on local changes of gene activity in *PXY*-positive stem cells, we introgressed fluorescent promoter reporters into our *pPXY:GR-ARF5ΔIII/IV* line and a line expressing *Myc-GR-bdl* under the control of the *PXY* promoter (*pPXY:Myc-GR-bdl*) and investigated reporter activity 1 day after Dex induction. As predicted by our transcriptional profiling (Supplementary Data 1), a *YFP* reporter driven by the cambium-associated *REDUCED IN LATERAL GROWTH1* (*RUL1*)[36] promoter (*pRUL1:YFP*) was more active in Dex-treated than in mock-treated *pPXY:GR-ARF5ΔIII/IV* lines (Fig. 5a, b, e, f) but repressed in Dex-treated *p35S:Myc-GR-bdl* plants (Fig. 5c, d, g, h). Instead, the *pWOX4:YFP* reporter displayed a reduced activity in *pPXY:GR-ARF5ΔIII/IV* and *pPXY:Myc-GR-bdl* plants upon Dex treatment (Fig. 5i–p) confirming our expectations. Of note, a reduction was only observed in cambium-borne stem cells but not in xylem parenchyma cells where *pWOX4:YFP* reporter activity was found as well, demonstrating that the effect was cambium-specific (Fig. 5i–p). In contrast to *RUL1* and *WOX4* promoters, *PXY* promoter activity visualized by the *pPXY:CFP* reporter did neither respond to Dex treatments of *pPXY:GR-ARF5ΔIII/IV* nor of *pPXY:Myc-GR-bdl* plants (Fig. 5q–x). These results demonstrated that our transcriptome analyses faithfully reported on gene activity in *PXY*-positive stem cells in response to different members of the auxin signalling machinery. Moreover, we concluded that there are distinct auxin response signatures of individual stem cell-related genes.

Because ARF5 induction resulted in both, *WOX4* repression and the induction of xylem- and phloem-related genes (Supplementary Fig. 8b), we reasoned that the repressive effect of ARF5 on cambium proliferation was due to an influence on the transition of cambial stem cells to vascular cells. To test this, we analysed the stem cell marker *pWOX4:YFP* in *mp-S319* mutants. Indeed, the radial extension of the *pWOX4:YFP* domain was increased in *mp-S319* plants (Fig. 6a–c), suggesting that the number of undifferentiated cambium cells was higher when *ARF5* activity was reduced. Consistently, when analysing the anatomy of the cambium zone predominantly the size of the domain of undifferentiated cells was increased (Fig. 6d–f), which was caused by an increase in cell number (Fig. 6g). This alteration resulted specifically in an increased ratio of undifferentiated cells to xylem cells (Fig. 6h). As in stems, increased cambium activity in *mp-S319* hypocotyls resulted mostly in an increase in phloem production but less in an increase in xylem production (Supplementary Fig. 7r-t). Likewise, Dex treatment of *pPXY:GR-ARF5ΔIII/IV* lines resulted in a decrease of the width of the cambium domain as well as the number of cambium cells. Moreover, predominantly the width of the phloem domain and the number of phloem cells was reduced (Fig. 7a, b). As in *mp-S319* mutants, this mostly affected the ratio of cambium/xylem cell numbers but to a lesser extent the cambium/phloem cell numbers (Fig. 7c, d). Taken together, these observations indicated that ARF5 predominantly fulfils its function by promoting the transition of undifferentiated stem cells to differentiated xylem cells.

**WOX4 mediates ARF5 activity**. Owing to the fact that *WOX4* was the only stem cell-associated gene being repressed by both GR-ARF5ΔIII/IV and Myc-GR-bdl induction and was identified as a mediator of auxin responses before[23], we hypothesized that ARF5 acts on cambium activity by directly regulating *WOX4*.

Indeed, the expression domain of the transcriptional *pWOX4:YFP* reporter[23] almost completely overlapped with *pARF5:mCherry* at the stem base (Fig. 8a–c). Moreover, *WOX4* transcript levels were increased in *mp-S319* mutant stems (Fig. 8d), indicating that the endogenous *ARF5* gene is required for the regulation of *WOX4*. Importantly, the negative effect of GR-ARF5ΔIII/IV induction on *WOX4* activity was also observed in the presence of the protein biosynthesis inhibitor cycloheximide (CHX) (Fig. 8e), which was in line with a direct regulation of *WOX4* by ARF5. Consistently, transient expression of ARF5ΔIII/IV in cultured cells had, similar as on other genes directly repressed by ARF5[50], a strong effect on the activity of a *pWOX4:LUC* promoter reporter, while this effect was only minor when ARF3 was expressed (Fig. 8f). This suggested that, in comparison to ARF3, ARF5 substantially influenced the activity of the *WOX4* promoter. Confirming this conclusion, neither cambium specific nor global induction of GR-ARF3 activity led to a significant change in *WOX4* expression in wild-type or *arf3;arf4* double mutants although *IPT3*, a putative downstream target of *ARF3*[51], was induced (Supplementary Fig. 9a-c). Furthermore, *WOX4* expression was not significantly altered in the *arf3;arf4* double mutant (Supplementary Fig. 9d), making it rather implausible that *ARF3* and *ARF4* act on cambium activity by regulating *WOX4*.

To confirm direct binding of the ARF5 protein to the *WOX4* promoter, we first predicted ARF-binding sites using a conventional weight matrix approach[52]. In total, we found two top-scored potential sites, located 335 bp (CAGACA) and 2175 bp (TGTCATtaCCGACA) upstream of the transcriptional start site (Supplementary Fig. 10). Performing electrophoretic mobility shift assays (EMSAs) using DNA oligomers covering sequences from those sites and sites shown to be bound by ARF5 before[50,53], binding was indeed observed for the first but not the second predicted site (Fig. 8g, h). These observations demonstrated a strong potential of the ARF5 protein to repress *WOX4* transcription via direct promoter binding.

To analyse the relevance of the observed effect of ARF5 on *WOX4* activity, we determined ICD extension in *mp-S319* and *wox4-1* single and double mutants. While *mp-S319* showed enhanced cambium activity (Fig. 9a, b, g), activity was similar in *wox4-1* single and in *mp-S319;wox4-1* double mutants (Fig. 9c, d, g), suggesting that *WOX4* is required for an ARF5-dependent repression of cambium activity. In comparison, depletion of *ARF5* activity in *mp-S319;pxy-4* double mutants lead to a mild suppression of cambium defects observed in *pxy-4* single mutants (Fig. 9e, f, g)[36], suggesting that the epistatic relationship between *WOX4* and *ARF5* is specific.

## Discussion

Similar to apical meristems, the regulation of the vascular cambium has been tightly associated with the plant hormone auxin for several decades[15,17,54]. However, spatial organization of functional signalling domains and the role of auxin signalling in controlling different aspects of cambium activity remained unknown. Here we show that auxin signalling takes place in cambium stem cells and that this signalling is crucial for cambium activity. We also show that not only stem cell activity in general but also the balance between undifferentiated and stem cells depends on the auxin signalling machinery with *ARF5* fulfilling a rather specific and *WOX4*-dependent role in this respect. Thus auxin-related signalling controls distinct aspects of cambium activity important for a dynamic tissue production and a complex growth process.

The concentration of IAA peaks in the centre of the cambial zone in *Populus* and *Pinus*[13–15] and transcriptional profilings indicated a spatial correlation of this peak with the expression of

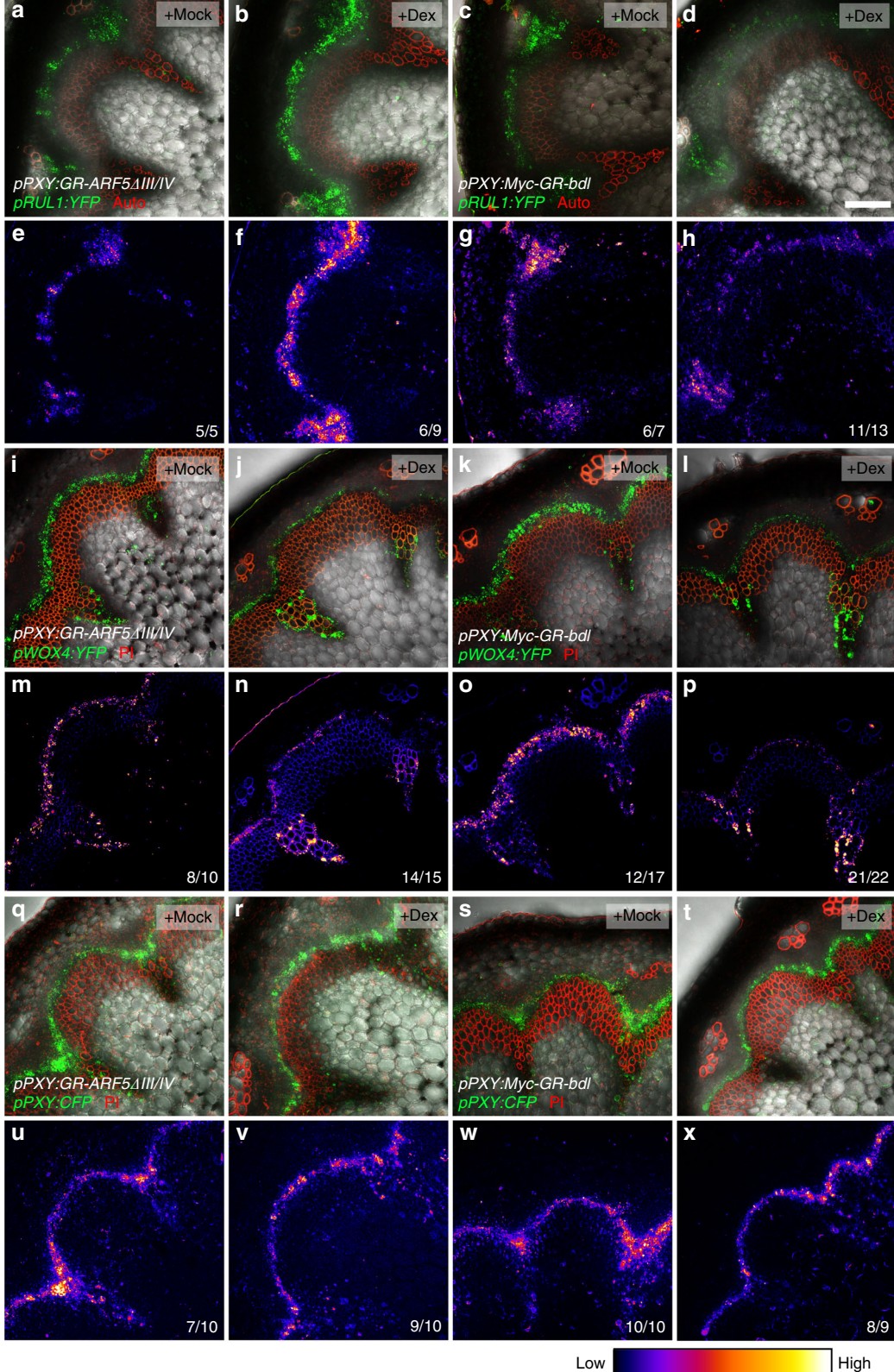

**Fig. 5** *ARF5* differentially regulates gene activity in cambium stem cells. **a–h** *pRUL1:YFP* activity in mock (**a**, **c**, **e**, **g**) and Dex (**b**, **d**, **f**, **h**)-treated *pPXY:GR-ARF5ΔIII/IV* (**a**, **b**, **e**, **f**) and *pPXY:Myc-GR-bdl* (**c**, **d**, **g**, **h**) lines. **i–p** *pWOX4:YFP* activity in mock (**i**, **k**, **m**, **o**) and Dex (**j**, **l**, **n**, **p**)-treated *pPXY:GR-ARF5ΔIII/IV* (**i**, **j**, **m**, **n**) and *pPXY:Myc-GR-bdl* (**k**, **l**, **o**, **p**) lines. **q–x** *pPXY:CFP* activity in mock (**q**, **s**, **u**, **w**) and Dex (**r**, **t**, **v**, **x**)-treated *pPXY:GR-ARF5ΔIII/IV* (**q**, **r**, **u**, **v**) and *pPXY:Myc-GR-bdl* (**s**, **t**, **w**, **x**) lines. **e–h**, **m–p** and **u–x** show false colour representations of relative intensity of reporter activity displayed in **a–d**, **i–l** and **q–t**, respectively. Colour coding is shown below. Numbers indicate the ratio of sample numbers showing the distinct effects among all samples analysed. Size bar in **d** represents 100 μm. Same magnification in all images

auxin signalling components[17,55]. However, genes responding to auxin were rather expressed in developing xylem cells arguing that sites of intense auxin signalling and of downstream responses do not necessarily overlap[18]. Consistently, our analysis of the highly sensitive auxin response marker *pDR5revV2:YFP* revealed a moderate auxin response in *PXY*-positive stem cells and a higher response in differentiated vascular tissues. Importantly, the auxin response in the *PXY*-positive region is overall pivotal for cambium activity since its local repression resulted in reduced tissue production similar as found in *wox4* or *pxy* mutants defective for canonical regulators of stem cell activity[23,28,36]. This demonstrates that, in the cambium, auxin signalling promotes stem cell activity in a cell-autonomous manner. Interestingly, *ARF5* and auxin signalling acts upstream of *WOX5* in the context of RAM organization[24,56] but differentiation of distal root stem

cells is promoted by *ARF10* and *ARF16*[56,57]. In comparison, *ARF5* restricts the stem cell domain in the SAM by repressing stem cell-related features[5]. In the RAM and the SAM, *ARF5* expression is found next to the expression domains of the central regulators *WOX5* and *WUS*, respectively[5,24,58], whereas it overlaps completely with the domain of *WOX4* expression in the cambium. Thus a division of labour of different auxin signalling components is found in various plant meristems and recruitment of distinct factors and adaptation of expression domains seem to have happened during the evolution of those systems.

*ARF5* plays a major role in translating auxin accumulation into the establishment of procambium identity in embryos and leaf primordia (reviewed in ref. [54]). However, ARF5 is also tightly associated with xylem formation via its direct targets *TMO5* and *ATHB8*[47,59,60]. In fact, we identified those and other xylem-associated genes like *ACAULIS5* (*ACL5*), *SUPPRESSOR OF ACAULIS5 LIKE3* (*SACL3*) and *BUSHY AND DWARF2* (*BUD2*)[21,60,61] to be induced upon *GR-ARF5ΔIII/IV* induction. Together with the observation that the domain of *WOX4*-positive cells is enlarged and the cambium/xylem ratio but not the cambium/phloem ratio is increased in *arf5* mutants, this suggests that *ARF5* promotes the transition of cambium stem cells to xylem cells (Fig. 9h). Because we also revealed a negative effect of *ARF5ΔIII/IV* induction on *WOX4* activity, a responsiveness of the *WOX4* promoter in transient expression systems, binding of the ARF5 protein to a distinct canonical AuxRE element in the WOX4 promoter and an epistatic genetic relationship between *WOX4* and *ARF5*, we propose that ARF5 fulfils its function partly by directly attenuating *WOX4* activity. Therefore, we propose that ARF5 acts as one hub modulating the activity of a multitude of genes in *PXY*-positive cells to foster the transition from stem cells to differentiated vascular cells.

Of note, one or several ARF transcription factors specifically promoting stem cell attributes in *PXY*-positive cells most likely exist because blocking auxin signalling or reducing auxin levels in

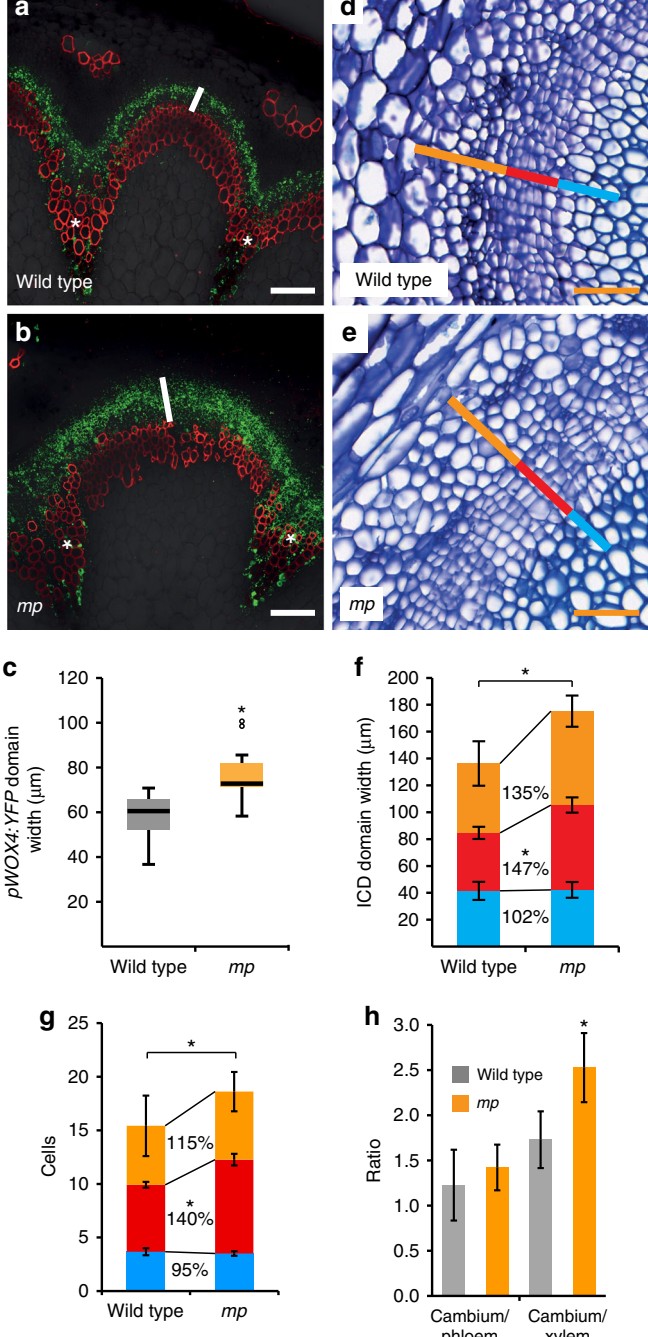

**Fig. 6** ARF5 activity reduces the number of cambium stem cells. **a**, **b** Confocal analyses of cross-sections at the stem base of wild-type and *mp-S319* mutant plants containing *pWOX4:YFP*. White bars mark the *pWOX4:YFP* domain. Asterisks mark vascular bundles. Size bars represent 100 μm. **c** Quantification of *pWOX4:YFP* domain width at the stem base of wild-type and *mp-S319* plants. Student's *T*-test was performed comparing wild-type and *mp-S319* mutants ($p = 2.65\text{E-}03$, sample size $n = 8–11$). The T-bars that extend from the boxes (whiskers) are expected to include 95% of the data. Significance is indicated by the asterisk. **d**, **e** Toluidine blue-stained cross-sections at the stem base of wild-type and *mp-S319* plants. Interfascicular regions are shown and the width of the vascular tissues phloem (orange bar), cambium (red bar) and xylem (blue bar) are marked. Size bars represent 50 μm. **f** Quantification of the width of the vascular tissues phloem (orange), cambium (red) and xylem (blue) and the ICD at the stem base of wild-type and *mp-S319* plants. Student's *T*-test was performed comparing wild-type and *mp-S319* mutants (phloem $p = 8.52\text{E-}02$, cambium $p = 2.67\text{E-}04$, xylem $p = 0.87$ and ICD $p = 3.06\text{E-}02$, sample size $n = 5–8$). Error bars represent ±standard deviation. Significance is indicated by asterisks. **g** Quantification of phloem (orange), cambium (red) and xylem (blue) cell numbers in interfascicular regions. Student's *T*-test was performed comparing wild-type and *mp-S319* mutants (phloem $p = 0.34$, cambium $p = 3.67\text{E-}05$, xylem $p = 0.57$ and all cells $p = 0.02$, sample size $n = 5–8$). Error bars represent ±standard deviation. Significance is indicated by asterisks. **h** Ratio of the radial extensions of cambium vs. phloem and cambium vs. xylem in wild-type and *mp-S319* mutants. Student's *T*-test (cambium/xylem $p = 1.09\text{E-}02$) or Welch's *T*-test (cambium/phloem $p = 1.0$) were performed comparing wild-type and *mp-S319* mutants (sample size $n = 5–8$). Error bars represent ±standard deviation. Significance is indicated by the asterisk

those cells counteract cambium activity overall (Fig. 9h). A scenario in which a concerted action of various ARFs regulate different aspects of cambium activity provides an explanation for why ARF5 acts as a transcriptional repressor of *WOX4* in a cambium context but *WOX4* depends positively on auxin signalling overall. Competition for promoter binding between ARF5 and ARFs with a stronger positive effect on WOX4 transcription would classify ARF5 as a transcriptional repressor in this specific case. Alternatively, ARF5-dependent recruitment of other transcriptional repressors, whether directly or indirectly, to the *WOX4* promoter would have the same effect. Especially our finding that ARF5 binds only to one out of the two predicted ARF-binding sites in in vitro assays gives room for a more complex occupancy of the *WOX4* promoter by ARF transcription factors in vivo. Consistent with the possibility that ARF5 does not necessarily act as a transcriptional activator, it represses *AUXIN RESPONSE REGULATOR15* (ARR15) and *STOMAGEN* in the SAM and in leaf mesophyll cells, respectively[5,50]. As observed for *WOX4*, both promoters are yet induced by ARF5 in transient expression systems[50]. The role of ARF5 in transcriptional regulation does therefore depend on the target promoter and the respective cellular environment, a phenomenon which has also been described for other ARF transcription factors[62,63]. Remarkably, not only xylem-related but also phloem-related genes are activated in stems when inducing *GR-ARF5ΔIII/IV* plants. This would argue for a general role of *ARF5* in promoting vascular differentiation and for the existence of one pool of stem cells marked by *PXY* promoter activity and feeding both xylem and phloem production. Alternatively, promotion of xylem differentiation is translated rapidly into the promotion of phloem differentiation by cell-to-cell signalling. The fact that the stem cell-to-phloem ratio is not altered in *arf5* mutants would argue for the latter option.

Consistent with a crucial role of cell-autonomous auxin signalling in cambium stem cells, *ARF3*, *ARF4* and *ARF5* expression was found in *PXY*-positive cells with *ARF5* being exclusively active in those. *ARF3* and *ARF4* have previously been shown to act in part redundantly in the establishment of abaxial identity in lateral organs[37]. In line with this function, we found both genes being mostly expressed distally of the cambium in phloem-related cell types. In fact, the lack of any effect on cambium or *WOX4* activity when modulating *ARF3* activity exclusively in *PXY*-positive cells suggests that at least *ARF3* functions outside of this domain when regulating cambium activity and that ARF transcription factors positively regulating *WOX4* transcription still have to be discovered. Whether the phloem-related expression of *ARF3* and *ARF4* modulates the activity of cambium regulators expressed in areas distally to the *PXY* expression domain like MORE LATERAL GROWTH1 (MOL1)[29] or CLAVATA/ESR41/42/44 (CLE41/42/44)[26,27,64] remains to be determined. Transcriptionally, at least, our modulation of auxin signalling had no effect on *MOL1* or *CLE41/42/44* mRNA abundance.

Collectively, we found a role of auxin signalling in the cambium sharing features with both the situation in the RAM where auxin regulates cell divisions[65] and the SAM where auxin, and particular ARF5, is strongly correlated with cell differentiation[4]. Thereby, we enlighten a long-observed role of auxin signalling in radial plant growth and reveal that its function is partly specific in different stem cell niches. The involvement of different auxin signalling components regulating individual aspects of meristem activity may provide a set-up required for regulating a complex developmental process by one signalling molecule.

## Methods

**Plant material**. All plant lines used in this study were *Arabidopsis thaliana* (L.) Heynh. plants of the accession Columbia (Col-0), except for the *mp-B4149* mutant,

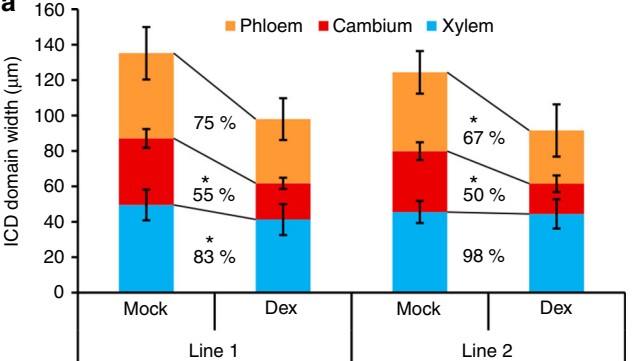

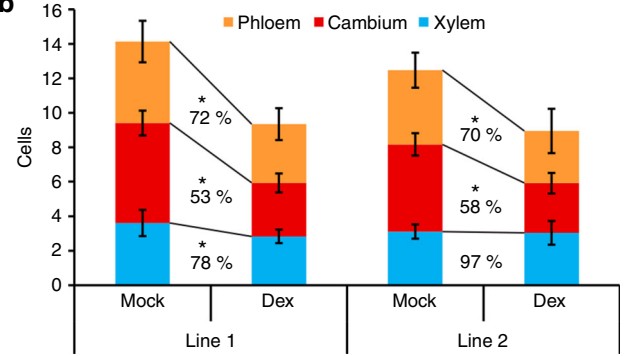

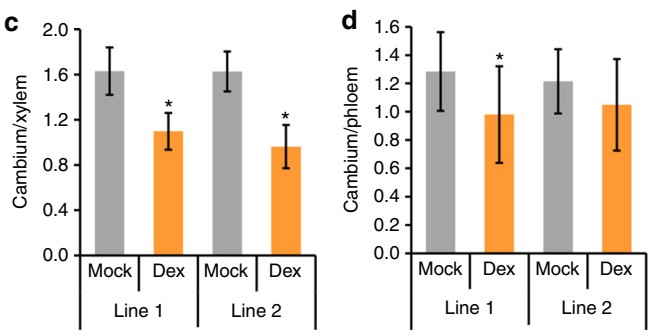

**Fig. 7** *ARF5* predominantly affects the cambium-to-xylem ratio. **a** Quantification of the width of the vascular tissues phloem (orange), cambium (red) and xylem (blue) and the ICD at the stem base of two mock and Dex-treated *pPXY:GR-ARF5ΔIII/IV* lines. Student's *T*-test was performed comparing mock and Dex-treated plants (line 1: phloem $p = 0.063$, cambium $p = 6.8E-08$, xylem $p = 0.49$; line 2: phloem $p = 0.036$, cambium $p = 1.28E-06$, xylem $p = 0.76$, sample size $n = 8–10$). Error bars represent ±standard deviation. Significance is indicated by the asterisk. **b** Quantification of cell numbers in interfascicular regions. Student's *T*-test was performed comparing two mock and Dex-treated *pPXY:GR-ARF5ΔIII/IV* lines (line 1: phloem $p = 0.013$, cambium $p = 1.98E-08$, xylem $p = 0.012$; line 2: phloem $p = 0.03$, cambium $p = 1.54E-06$, xylem $p = 0.79$, sample size $n = 8–10$). Error bars represent ±standard deviation. Significance is indicated by the asterisk. **c** Ratio of cell numbers in cambium vs. xylem in two mock and Dex-treated *pPXY:GR-ARF5ΔIII/IV* lines. Student's *T*-test was performed comparing mock and Dex-treated lines (line 1: $p = 5.62E-06$, line 2: $p = 1E-06$). Error bars represent ±standard deviation. Significance is indicated by the asterisk. **d** Ratio of cell numbers in cambium vs. phloem in two mock and Dex-treated *pPXY: GR-ARF5ΔIII/IV* lines. Student's *T*-test was performed comparing mock and Dex-treated lines (line 1: $p = 0.042$, line 2: $p = 0.22$). sample sizes $n = 8-10$. Error bars represent ±standard deviation. Significance is indicated by asterisks

which has the Utrecht background[41,66]. The *arf3-1* (SAIL_1211_F06, N878509), *ett-13* (SALK_040513, N540513), *mp-S319* (SALK_021319, N521319), *wox4-1* (GK_462GO1, N376572) and *pxy-4* (SALK_009542, N800038) mutants, as well as the *pDR5rev:GFP* reporter line (N9361[30]), were ordered from the Nottingham Arabidopsis Stock Centre (NASC). The *mp-B4149* and *arf4-2* (SALK_070506) mutant were provided by Dolf Weijers (University of Wageningen, The

Netherlands) and Alexis Maizel (COS Heidelberg, Germany), respectively. Genotyping was performed by PCR using primers listed in Supplementary Table 1. Genotyping of *mp-B4149* was done using the MP_for8/MP_rev8 primer pair for amplification and the MseI restriction enzyme for detecting polymorphisms.

**Plant growth and histological analyses**. Plants destined for histology were grown for 3 weeks in short day (SD) conditions (8 h light and 16 h dark) and then transferred to long day (LD) conditions (16 h light and 8 h dark) to induce flowering and used for histology at a height of 15–20 cm. Stem segments of at least 1 cm in length (including the stem base) were harvested, embedded in paraffin and sectioned using a microtome (10 μm sections). After deparaffinization, sections were stained with 0.05% toluidine blue (Applichem), fixed with Entellan (Merck) and imaged using a Pannoramic SCAN digital slide scanner (3DHistech). Pictures were analysed in a blind test using the Pannoramic Viewer 1.15.4 software (3DHistech). For quantitative analyses, at least five plants were analysed for each data point.

**Statistical analyses**. Statistical analyses were performed using IBM SPSS Statistics for Windows, Version 21.0. Armonk, NY: IBM Corp. Means were calculated from measurements with sample sizes as indicated in the respective figure legends. In general, all displayed data represent at least two independent, technical repetitions, unlike otherwise indicated. Error bars represent ±standard deviation. All analysed datasets were prior tested for normal distribution by Kolmogorov–Smirnov test and for homogeneity of variances by the Levene statistic. Significant differences between two datasets were calculated by applying a Welch's *t*-tests or Student's *T*-test depending on the homogeneity of variances. The significance thresholds were set to *p*-value <0.05 (indicated by one asterisk). For multiple comparisons between three or more datasets, a one-way analysis of variance was performed, using a confidence interval of 95% and a post-hoc Bonferroni for comparisons of data sets of homogenous variances or a post-hoc Tamhane-T2 in case variances were not homogenous.

**Sterile culture**. To induce adventitious root formation in the strong *arf5* mutant allele *mp-B4149*, seeds were liquid sterilized by 70% ethanol and incubation in 5% sodiumhyperchloride followed by three washes with ddH$_2$O. After stratification at 4 °C in the dark for 3 days, seeds were sown on 1/2 Murashige–Skoog (MS) medium plates (including B5 vitamins) in rows and grown vertically. After 7 days of growth under SD (8 h light, 16 h dark) conditions, rootless mutant as well as wild-type-looking seedlings from the segregating population were bisected with a scalpel and transferred to adventitious root-inducing medium (1/2 MS (including B5 Vitamins) + 1.5 % sucrose + 3 μg/ml indole butyric acid + 0.7 % agar + 50 μg/ml ampicillin)[43]. After additional 2 weeks of growth under SD conditions, successfully rooted seedlings were transferred to soil and grown under SD conditions for 1 additional week before they were transferred to LD conditions. Plants that survived the transfer were genotyped for *mp-B4149* and only wild-type and

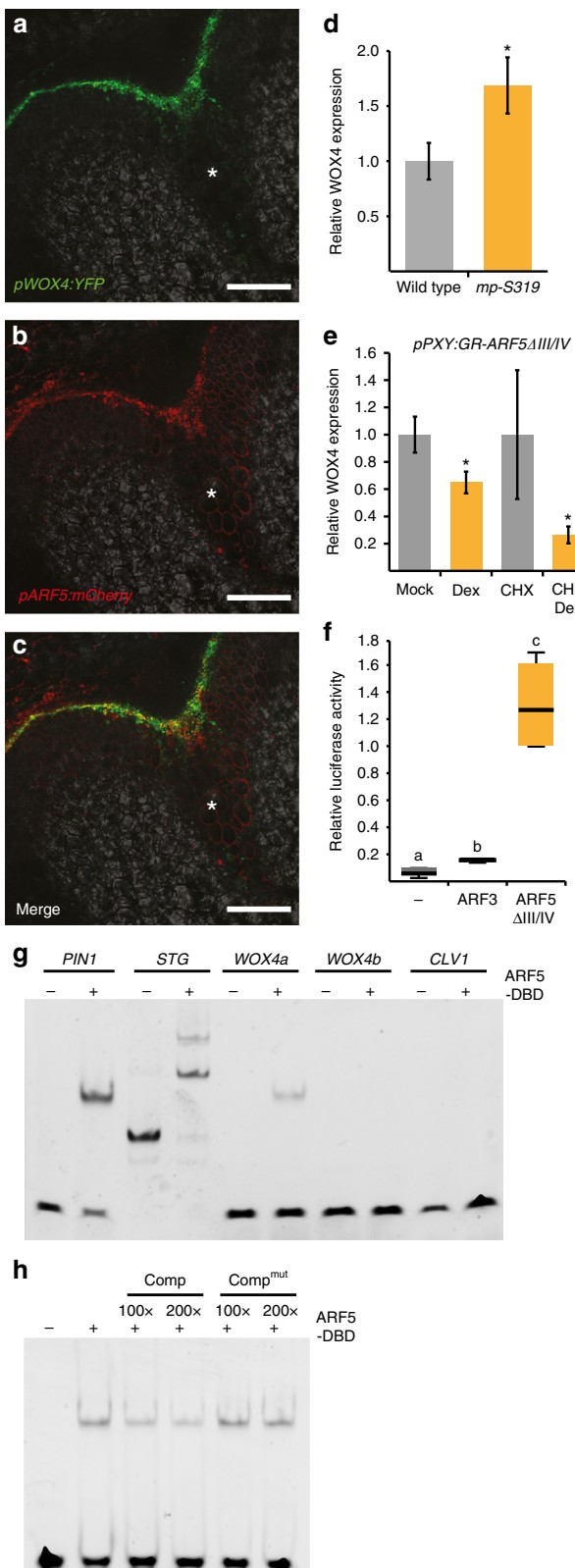

**Fig. 8** : ARF5 binds to the *WOX4* promoter. **a–c** Confocal analysis of stem bases of plants carrying the *pWOX4:YFP* and the *pARF5:mCherry* reporter. Asterisks mark the vascular bundles. Size bars represent 100 μm. **d, e** Analysis of *WOX4* transcript levels by quantitative RT-PCR at the stem base of wild-type and *mp-S319* mutant pants (**d**) and in the second internode of *pPXY:GR-ARF5ΔIII/IV* plants after mock or Dex and CHX or CHX/Dex (**e**) treatments. Student's *T*-test (wild type and *mp-S319* *p* = 1.74E-02 and Mock and Dex *p* = 1.66E-02) or Welch's *T*-test (CHX and CHX/Dex *p* = 1.29E-02) were performed comparing wild-type and *mp-S319* mutants, mock and Dex and CHX and CHX/Dex, respectively (Sample sizes *n* = 3–6). Error bars represent ±standard deviation. Significance is indicated by the asterisk. **f** Analysis of relative luciferase activity of a *pWOX4:LUC (firefly);p35S:LUC (Renilla)* reporter in *Arabidopsis* protoplasts in the presence of *p35S:ARF3*, *p35S:ARF5ΔIII/IV* or no effector construct. Relative luciferase activity was determined by dual-luciferase assays. Statistical groups indicated by letters were determined by one-way ANOVA with post hoc Tamhane-T2 (CI 95%, Sample size *n* = 4–5). The T-bars that extend from the boxes (whiskers) are expected to include 95% of the data. **g** EMSAs probing DNA oligomers with proven (*PIN1, STOMAGEN*)[50,53] or predicted (*WOX4*) binding capacities by the ARF5 DNA-binding domain (ARF5-DBD). An oligomer from the *CLAVATA1* (*CLV1*) promoter was used as a negative control. Oligomer sequences are shown in Supplementary Table 1. **h** EMSAs probing the *WOX4a* oligomer in the presence of the non-labelled *WOX4a* competitor (comp) (in 100× and 200× excess) and in the presence of a mutated *WOX4a* oligomer (comp$^{mut}$) (with 100× and 200× excess)

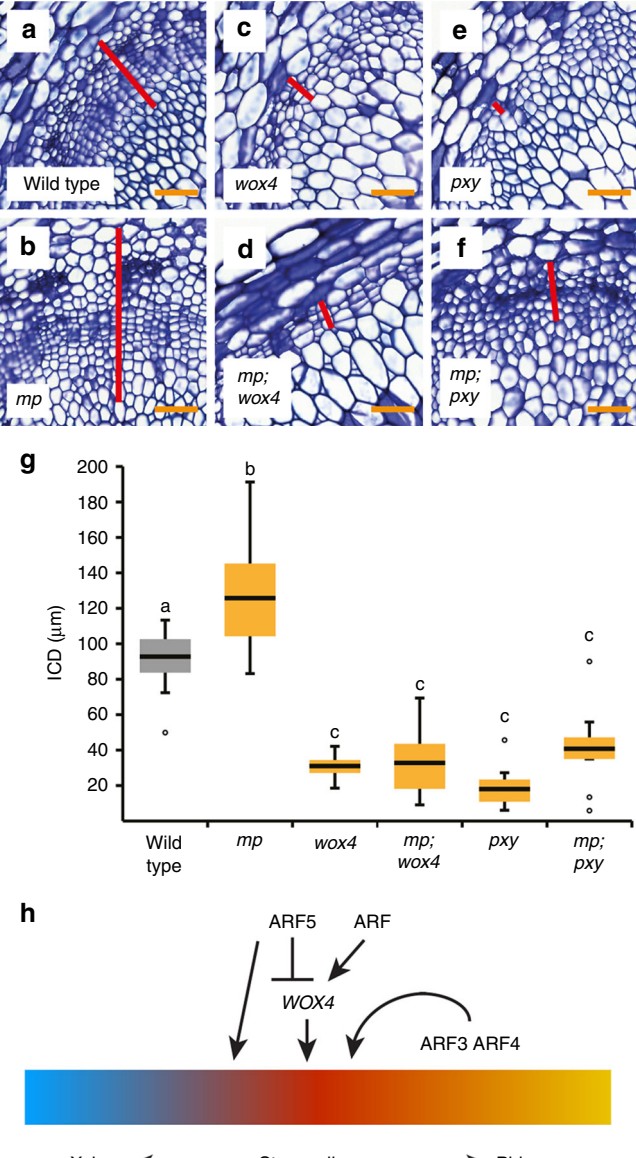

Fig. 9 WOX4 is required for ARF5 activity. **a–f** Toluidine blue-stained cross-sections at the stem base of wild-type and single and multiple mutant plants. Genotypes are indicated. Interfascicular regions are shown and ICD tissues are marked (red bar). Size bars represent 50 μm. **g** Quantification of ICD tissue extension at the stem base of plants shown in **a–f**. Statistical groups indicated by letters were determined by one-way ANOVA with post hoc Tamhane-T2 (CI 95%, Sample size $n = 9$–12). The T-bars that extend from the boxes (whiskers) are expected to include 95% of the data. **h** Scheme representing the proposed effect of ARF transcription factors on cambium activity. We suggest that ARF5 predominantly promotes xylem production by directly activating xylem-related genes and by repressing WOX4. Other ARFs, which still have to be identified, may activate WOX4 at the same time. ARF3 and ARF4 act positively on cambium activity from outside of the stem cell domain and independently from WOX4

homozygous mutant plants were used for histological analysis at a plant height of 15–20 cm as described in the previous section.

**Plasmid construction**. To avoid diffusion, all fluorescent proteins were targeted to the ER by fusing them to the corresponding sequence motif (ER+HDEL motif[67]). For generating pDR5revV2:YFP (pKB46), we initially inserted the ADAPTOR PROTEIN-4 MU-ADAPTIN (AP4M, At4g24550) terminator, amplified from genomic DNA using the At4g24550_for1/At4g24550_rev1 primer pair, into pLC075 containing the DR5revV2 promoter fragment[31] using BamHI/XhoI

restriction sites. A fragment carrying the ER-EYFP-HDEL coding sequence (CDS) was inserted in the resulting pLC075:AP4Mterm using the BamHI restriction site, to obtain pLC075:YFP:AP4Mterm. The complete reporter fragment was inserted in the binary vector pGreenII0179[68] using KpnI/XhoI restriction sites. For generating p35S:Myc-GR-bdl (pKB9), the Myc-GR-bdl fragment was amplified from genomic DNA of pRPS5a:Myc-GR-bdl[58] using the Myc_for1/BDL_rev3 primer pair. The resulting fragment was inserted into the pGreen0229 vector[68] containing the 35S promoter (pGreen0229-35S) using XbaI/BamHI restriction sites. To produce pAlcA:iaaM (pKB2), the iaaM CDS was amplified from piaaM (pIND:IND-iaaM)[69] using the IAAMfor1/IAAMrev1 primer pair and introduced into pGreen0229-AlcA[32] using AatII/EcoRI restriction sites. pWOX4:AlcR (pTOM55) was produced by amplifying the WOX4 promoter using the primers WOX4for11/WOX4ref9 and inserting the resulting fragment into pAlcR-GUS[32] using SpeI/NotI sites. The pPXY:Myc-GR-bdl (pKB45) construct was generated by cloning the Myc-GR-bdl fragment, amplified from pKB9 using the Myc_for5/BDL_rev7 primer pair, in pGreen0229 containing the PXY promoter (pTOM50[28]) using NcoI/Cfr9I restriction sites. The promoter regions of BDL[35] were amplified from genomic DNA using the BDL_for2/BDL_rev4 and BDL_for3/BDLrev5 primer pairs. Both fragments were cloned into pGreenII0179 using NotI/XbaI and Cfr9I/KpnI restriction sites, respectively. The resulting plasmid (pKB27) was used to produce the pBDL:YFP (pKB28 using ER-EYFP-HDEL) and pBDL:Myc-GR-bdl (pKB29) constructs by inserting fragments carrying the respective CDSs using NcoI/Cfr9I restriction sites. For generating ARF3, ARF4 and ARF5 reporter constructs, promoter regions of the three genes were amplified from genomic DNA using the ARF3for1/ARF3rev1 and ARF3for2/ARF3rev2, ARF4for1/ARF4rev1 and ARF4for2/ARF4rev2 and MP_for7/MP_rev5 and MP_for5/MP_rev6 primer pairs. Both fragments were cloned for each gene into pGreen0229 using NotI/BamHI and BamHI/KpnI (ARF3), NotI/SpeI and Cfr9I/KpnI (ARF4) and NotI/BamHI and BamHI/ApaI (ARF5) restriction sites. The resulting plasmids (pKG40 (ARF3), pKG41 (ARF4) and pKB3 (ARF5)) were used to produce the pARF3:YFP (pKB30), pARF4:YFP (pKB31), pARF5:YFP (pKB24) and pARF5:mCherry (pKB4) constructs by inserting fragments carrying the respective CDSs using BamHI, SpeI and NdeI/XhoI restriction sites, respectively. To produce p35S:GR-ARF3 (pKB42) and p35S:GR-ARF5 (pKB17), we amplified the GR open reading frame from pKB9 using the GR_for1/GR_rev3 primer pair and inserted the resulting fragment in pGreen0229-35S using XbaI/Cfr9I restriction sites. An additional unannotated SalI restriction site in the pGreen0229 backbone was removed by PCR-based silent mutagenesis using the NOS-mut_for1/NOS_mut_rev1 primer pair. In the resulting pGreen0229-35S:GR (pKB41) vector, we inserted the ARF3 and ARF5 CDS, amplified from cDNA using the ARF3_for4/ARF3_rev4 and MP_for16/MP_rev14 primer pair, using SalI/Cfr9I and SalI/EcoRI restriction sites, respectively. pPXY:GR-ARF3 (pKB43) was generated by cloning the GR-ARF3 fragment, amplified from pKB42 using the GR_for5/ARF3_rev4 primer pair, in pTOM50 using NcoI/Cfr9I restriction sites. For generating pPXY:GR-ARF5ΔIII/IV (pKB25), the GR-ARF5ΔIII/IV fragment with a stop codon was amplified from pKB17 using the MP_for18/MP_rev16 primer pair and inserted in pTOM50 using NcoI/Cfr9I restriction sites. For generating the pWOX4:LUC (firefly);p35S:LUC (Renilla) (pKB55) reporter, the pZm3918:LUC (firefly) fragment in pZm3918:LUC (firefly);p35S:LUC (Renilla) (pGreen-LUC-REN) was excised by digest with KpnI/XbaI and replaced by a pWOX4:LUC (firefly) fragment previously excised from pWOX4:LUC (pMS80) using KpnI/NheI restriction sites. To produce p35S:ARF3 (pKB44), the ARF3 CDS was amplified from pKB42 using the ARF3_for6/ARF3_rev4 primer pair and introduced it in pGreen0229-35S using XbaI/Cfr9I restriction sites. For generating p35S:ARF5ΔIII/IV (pKB40), the ARF5ΔIII/IV CDS was amplified from pKB25 using MP_for17/MPrev15 and introduced in pGreen0229-35S using XbaI/EcoRI restriction sites. All constructs were sequenced, and after plant transformation by floral dip[70], single copy transgenic lines were identified by Southern blot analyses and representative lines were used for crosses and further analyses. The promoter regions of RUL1[36] were amplified from genomic DNA using the RULfor2/RULrev2 and RULfor3/RULrev3 primer pairs. Both fragments were cloned into pGreen0229[68] using XhoI/EcoRI and SpeI/NotI restriction sites, respectively. The resulting plasmid (pJA16) was used to produce the pRUL1:YFP (pJA19 using ER-EYFP-HDEL) construct by inserting fragments carrying the respective CDS using NdeI/XmaI restriction sites. For generation of the bacterial expression vector pMG210, a 6xHis tagged N-terminal fragment of the maltose binding protein (MBP)[71] was amplified from pMAL-c2X (New England Biosciences) using the malE_fw2/malE_rv1 primer pair and inserted into pMAL-c5X (New England Biosciences) using MunI/BglII restriction sites. The plasmid pMG210 was further modified by replacing the C-terminal fragment of MBP, which contains the Factor Xa protease cleavage site, with the respective fragment from pETMBP_1a[72], which contains a TEV protease cleavage site, by using BglII/NcoI restriction sites to generate pJH036. To produce the final expression vector pMG307, the ARF5 DNA-binding domain (ARF5-DBD, aa 120-274) was amplified from cDNA using the ARF5-DBD_fw1/ARF5-DBD_rv1 primer pair and cloned into pJH036 using NcoI/XhoI and NcoI/SalI restriction sites, respectively. All primers mentioned in this section are listed in Supplementary Table 1.

**Confocal microscopy and image analysis**. For imaging fluorescent reporter lines in stems, rough hand sections were produced with a razor blade (Wilkinson Sword) and analysed using an LSM 780 spectral confocal microscope (Carl Zeiss) equipped

with the Zen 2012 software (Carl Zeiss) and a Nikon A1+ confocal microscope with the Nikon NIS Elements AR4.50.00 software. Stem sections (except for *pARF5:mCherry*) were counterstained for 5 min with 5 μg/ml propidium iodide (PI; Merck) dissolved in tap water. PI was excited at 561 nm (DPSS laser) and detected at 590–690 nm. YFP was analysed with excitation at 514 nm (Argon laser) and detection at 516–539 nm. Cyan fluorescent protein (CFP) was excited at 458 nm (Argon laser) and detected at 462–490 nm, while green fluorescent protein (GFP) was excited at 488 nm (Argon laser) and detected at 499–544 nm. mCherry reporter activity was analysed with excitation at 561 nm (DPSS laser) and detection at 597–620 nm. Transmitted light pictures were generated using the transmission photo multiplier detector of the microscope. For Nikon A1+ confocal microscopy, CFP was excited with 457 nm and detected using the 482/35 filter. YFP was excited at 514 nm and detected with the 540/30 filter, PI was excited at 561 nm and detected with 595/50 filter, while autofluorescence of xylem cells were excited at 405 nm and detected with the 450/50 filter. Five-day-old *Arabidopsis* seedlings were counterstained with the cell membrane dye FM® 4–64 (Thermo Fisher Scientific) to visualize cell borders[73]. FM® 4–64 was excited at 561 nm (DPSS laser) and detected at 653–740 nm. For heat maps displaying reporter activity, images obtained using the fluorescence-specific channel from mock and Dex-treated samples were combined in one picture and analysed using ImageJ2. Fire mode in Lookup Tables shows the fluorescent signal's intensity as heat maps.

**Pharmacological treatments**. Stock solutions of 25 mM Dex (VWR) dissolved in 100% Ethanol and 10 mM CHX (Carl Roth) dissolved in ddH$_2$O were freshly prepared prior to use. For long-term Dex treatments, plants were initially grown for 3 weeks without treatment in SD conditions to circumvent growth defects during early plant development. Plants were then transferred to LD conditions and watered twice a week with either 15 μM Dex (25 mM Dex stock diluted in tap water) or mock solution (equal amount of 100% Ethanol in tap water) until they reached a height of 15–20 cm and were harvested for histology. For short-term Dex treatments 15–20 cm tall plants were dipped headfirst in 15 μM Dex (25 mM Dex stock diluted in tap water + 0.02% Silwet) or Mock solution (equal amount of 100% Ethanol in tap water + 0.02% Silwet) for 30 s. Subsequently, plants were transferred to LD growth conditions, watered with 15 μM Dex or Mock solution and incubated until harvest of second internodes for RNA isolation. For additional short-term CHX treatment, 10 mM CHX stock was added to the 15 μM Dex and Mock solution to a final concentration of 10 μM and the plants were treated in the same way as described before. For inducing the AlcA/AlcR system[32], plants where grown for 3 weeks in SD, transferred to LD until bolting. When plants where 0.5–3 cm tall, they were put under a plastic dome together with 2× 15 ml 70% ethanol in round petri dishes and left overnight. Plants were harvested 10 days after induction and wild-type plants were around 25 cm tall.

**Transient reporter activity assays**. For transient reporter activity assays, protoplasts derived from an *Arabidopsis* (Col-0) dark-grown root cell suspension culture (kindly provided by Claudia Jonak, GMI, Vienna) were isolated and transfected. For transfection, we used 10 μg of reporter construct (*pKB55*) containing *p35S:LUC* (Renilla) as an internal control and 10 μg of each effector construct. The transfected protoplasts were diluted with 240 mM CaCl$_2$ (1:3) followed by cell lysis and dual-luciferase assay using the Dual-Luciferase Reporter Assay System (Promega) and following the manufacturer's instructions. Luminescence was measured using a Synergy H4 Hybrid Multiplate Reader (BioTek). For each reporter/effector combination, 3–5 technical replicates were done and the experiments were repeated at least three times. For experimental analysis, Firefly Luciferase activity was normalized to Renilla Luciferase activity.

**RNA preparation and quantitative reverse transcriptase-PCR (qRT-PCR)**. Frozen plant material from second internodes or the stem base (including 5 mm above) of 15–20 cm tall plants (three biological replicates (three plants each) per genotype/treatment) were pulverized with pestle and mortar and RNA was isolated by phenol/chloroform extraction. RNA elution in RNase-free water was followed by treatment with RNase-free DNase (Thermo Fisher Scientific) and reverse transcription (RevertAid First Strand cDNA Synthesis Kit; Thermo Fisher Scientific). cDNA was diluted 1:25 prior to amplification. qRT-PCR was performed using SensiMix™ SYBR® Green (Bioline Reagents Ltd) mastermix and gene-specific primers (listed in Supplementary Table 1) in a Roche Lightcycler480 following the manufacturer's instructions. Experiments were performed in triplicates with plant material of three plants being pooled for each replicate. Two reference genes (ACT2 and EIF4a) were used to normalize our signal. Error bars: ±standard deviation. Raw amplification data were exported and further analysis and statistical tests were done using Microsoft Excel 2010.

**Transcriptional profiling**. Ten μg of total RNA for each sample were treated with RNase-free DNase (Thermo Fisher Scientific) and purified using RNA-MiniElute columns (Qiagen) following the manufacturer's protocol. Library preparation and next-generation-sequencing (NGS) was performed at the Campus Science Support Facilities (CSF) NGS Unit (www.csf.ac.at) using HiSeqV4 (Illumina) with single-end 50-nucleotide reads. Reads were aligned to the *A. thaliana* Columbia (TAIR10) genome using CLC Genomics Workbench v7.0.3 and analysed using the DESeq

package from the R/Bioconductor software[75]. Dex-treated samples were compared to mock-treated samples with a stringency of *p*-value <0.05 determined analogous to Fisher's Exact Test. Data processing was further analysed using VirtualPlant 1.3[76] Gene Sect and BioMaps with a cut-off *p*-value <0.05 and cut-off *p*-value <0.01, respectively, which was determined by Fisher's Exact Test (with false discovery rate correction). Data were aligned to The *Arabidopsis* Information Resource (TAIR) databases and as background population for all analysis the *A. thaliana* Columbia (TAIR10) genome was used. Further data processing was done in Microsoft Excel 2010.

**ARF-binding site prediction**. Three thousand bp upstream of the start codon of the *WOX4* gene were used for the ARF-binding site prediction. In this analysis, one frequency matrix from CIS-BP[77] (M0147_1.02) and two from a Dap-seq analysis[78] (ARF_ARF2_col_v31, ARF_ecoli_MP_col) were applied. We computed position weight matrices as log-odds weights[52]. The significance of potential sites was estimated as a *p*-value derived from a matrix score[79].

**ARF5-DBD purification from bacteria**. For the *Escherichia coli* expression culture 100 ml LB medium containing ampicillin (100 μg/ml) and chloramphenicol (25 μg/ml) was inoculated with 5 ml of an overnight culture of the Rosetta (DE3) pLysS strain (Novagen) previously transformed with *pMG307* and grown at 37 °C until an OD$_{600}$ of 0.6 was reached. Expression of the (6xHis)MBP-ARF5-DBD fusion protein was induced by adding IPTG to a final concentration of 1 mM, and the culture was incubated for additional 5 h. Cells were harvested by centrifugation at 3000 × *g* for 20 min and the pellet stored at −20 °C. All subsequent steps were performed on ice or at 4 °C. The recombinant fusion protein was released from thawed cells by resuspension of the pellet in 2 ml lysis buffer (20 mM Tris-HCl pH 8, 150 mM NaCl, 2 mM dithiothreitol (DTT), 0.1% Lysozyme (Roth), 7.5 U/ml Benzonase (Novagen), 1× complete EDTA-free Protease Inhibitor (Roche), 0.2% Nonidet P40) and pulsing three times for 10 s at 5% amplitude with the S-4000 sonicator (Misonix) using a microtip. The lysate was cleared by centrifugation at 10,000 × *g* for 20 min and the supernatant loaded on a pre-equilibrated Ni-NTA purification column. To prepare the Ni-NTA purification column, 400 μl of Ni-NTA Superflow resin (Qiagen) was applied to a Micro Bio-Spin Column (Bio-Rad), washed with 2 ml Milli-Q water (Micropore) and equilibrated with 2 ml wash buffer 1 (20 mM Tris-HCl pH 8, 150 mM NaCl, 2 mM DTT, 0.2% Nonidet P40, 10 mM Imidazole pH 8) using gravity flow. The loading step was repeated twice to enhance binding of the (6xHis)MBP-ARF5-DBD fusion protein. The resin was sequentially washed with 2 ml of each of the following buffers: wash buffer 1, wash buffer 2 (20 mM Tris-HCl pH 8, 150 mM NaCl, 2 mM DTT, 10 mM Imidazole pH 8), wash buffer 3 (20 mM Tris-HCl pH 8, 1 M NaCl, 2 mM DTT, 10 mM Imidazole pH 8), and wash buffer 4 (20 mM Tris-HCl pH 8, 1 M NaCl, 2 mM DTT, 20 mM Imidazole pH 8). The fusion protein was released from the resin by adding 1 ml elution buffer (20 mM Tris-HCl pH 8, 1 M NaCl, 2 mM DTT, 330 mM Imidazole pH 8), incubated with 10 μg TEV protease overnight and the reaction was subsequently loaded on a pre-equilibrated SP purification column. Preparation of the SP purification column was done by applying 400 μl of SP Sepharose Fast Flow (GE Healthcare) to a Micro Bio-Spin Column (Bio-Rad), washing with 2 ml Milli-Q water (Micropore) and equilibrating with 2 ml SP wash buffer (20 mM Tris-HCl pH 8, 150 mM NaCl, 2 mM DTT) using gravity flow. The loading step was repeated twice to enhance binding of the ARF5-DBD. The resin was washed by adding 2 ml of SP wash buffer, and the purified protein was released upon addition of 250 μl SP elution buffer (20 mM Tris-HCl pH 8, 500 mM NaCl, 2 mM DTT). Concentration of the purified ARF5-DBD was determined by absorption at 280 nm considering molecular weight and molar extinction coefficient and purity verified by sodium dodecyl sulfate-polyacrylamide gel electrophoresis followed by Coomassie Brilliant Blue staining.

**Fluorescent EMSAs**. EMSAs were performed using fluorescent probes produced by annealing complementary pairs of CY5 5′-labelled oligonucleotides (Eurofins) in annealing buffer (10 mM Tris-HCl pH 8, 1 mM EDTA, 50 mM NaCl). Oligonucleotide sequences are listed in Supplementary Table 1. For the binding reactions, 12.5 pmol of the ARF5-DBD was incubated with 200 fmol CY5 probe in binding buffer (final reaction conditions: 10 mM Tris-HCl pH 7.4, 75 mM NaCl, 50 ng/μl Poly(dI-dC) (Thermo Scientific), 1 mM DTT) for 20 min at room temperature. Samples were mixed with 6× Orange G loading dye and applied to a 6% native polyacrylamide gel in 0.5× TBE buffer. Electrophoresis was conducted at 300 V (~20 mA per gel) for 20 min and the gel shifts were subsequently recorded by using the Advanced Fluorescence Imager (Intas) and the ChemoStar Professional software with red dye filter settings (628/32–716/40). Competitor assays were performed as described above but protein amount was doubled and binding reactions were pre-incubated for 10 min with the non-labelled competitor before adding the fluorescent probe.

**Data availability**. Raw sequencing data produced in this study have been uploaded to NCBI's Gene Expression Omnibus (GEO) database[74] and are accessible through GEO Series accession number GSE98193. The authors declare that all other data supporting the findings of this study are available within the paper and its supplementary information files.

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

## Acknowledgements

This work was supported by the SFB 873 of the German Research Foundation (DFG) and a Heisenberg professorship to T.G. (DFG, GR2104/5-1). *arf4-2* mutants and the *pRPS5a: Myc-GR-bdl* line were kindly provided by Alexis Maizel (COS Heidelberg, Germany) and Gerd Jürgens (University of Tübingen, Germany), respectively. *mp-B4149* mutant seeds and the *pLC075* construct were obtained from Dolf Weijers (University of Wageningen, The Netherlands). Armin Djamei (GMI, Austria) donated the *pGreen-LUC-REN* construct. An established *Arabidopsis* (Col-0) dark-grown root cell suspension culture was a kind gift from Claudia Jonak (GMI, Austria). Jana Hakenjos (COS, Heidelberg, Germany) contributed to the EMSAs. The work of V.G.L. and D.N. was supported by the Russian Science Foundation (17-74-10102). D.N. acknowledges Wageningen University for a Sandwich PhD scholarship. We thank members of the Greb laboratory for helpful discussions on the manuscript.

## Author contributions

K.B., J.Q., M.G. and T.G. conceived and designed the experiments. K.B., J.Q., M.G., V.J., T.S., K.G., E.-S.W., D.D.N., V.G.L., J.A. and P.S. performed experiments and analysed data. J.U.L. and T.G. analysed data. K.B. and T.G. wrote the manuscript.

## Additional information

**Competing interests:** The authors declare no competing financial interests.

**Reprints and permission** information is available online at http://npg.nature.com/ reprintsandpermissions/

