## [Peer Review File · Nature Communications]

Reviewers' comments:

Reviewer #1 (Remarks to the Author):

In the manuscript by Brackmann and co-workers, the authors describe the spatial specificity of auxin responses during secondary growth in Arabidopsis. More specifically, they have uncovered that the well-known Auxin Response Factor ARF5/MOMOPTEROS represses WOX4 thereby restricting cambium activity. They also show that two other ARFs act as activators in tissues outside of this domain marked by WOX4 expressing cells. Hence, the authors connect several well-known pathways into the context of secondary growth and highlight the tissue specific activity of these factors. Similar to other work by this group, experiments are thoroughly worked out; all required controls are present and figures are of high quality. I do however have a few matters I would like the authors to address.

Using transcriptomics, the authors pinpoint WOX4 as a putative target to continue working on. It is however not entirely clear why this factor (which is well studied in the author's lab) is chosen and not one of the other known and interesting genes. Related to this, I found very little confirmation for this transcriptomics study. A second issue here is the fact that entire stem segments were taken. Although this is not a problem per se, it does become an issue when statements are made about the tissue specificity of factors. It is very plausible that a factor is induced in one tissue upon ectopic induction with GR, while it is repressed in another tissue. This would have implications in classifying genes into induced and repressed etc. In order to solve this (including the previous comment on confirmation), the authors should take a few marker lines and check that these are indeed induced or repressed in the respective tissues as is assumed at the moment. Many marker lines of genes in the lists are available in the community.

As another comment related to this paragraph, it is not very clear to me at the moment why the authors propose ARF5 to function by promoting the transition of undifferentiated cells to differentiated xylem. Could the authors try to clarify this statement better with the evidence at hand, as it is one of the main conclusions in the manuscript. The similar statement is written down in the discussion section where it is stated that ARF5 promotes transition of stem cells to xylem cells based on the fact that genes such as ACL5, SACL3 and ATHB8 are recovered. In my opinion, an alternative hypothesis is that these genes are found simply because they are auxin dependent and this part of the auxin signaling pathway goes through ARF5 (as has been shown in literature). Hence, I don't see how finding these genes contributes to the statement that is made.

As a final comment to this section, it would also help to better explain or hypothesize how and why pPXY:GR-ARF5 Δ III/IV and p35S:Myc-GR-bdl can have similar effects on gene expression upon induction with DEX (this could be related to the comment above about tissue specificity of the transcriptomics experiment when taking whole stem segments); and why ARF5 would act as transcriptional repressor as it is generally considered to be an activator ARF.

In the paragraph about the link between WOX4 and ARF5, the authors performed all experiments possible to show that WOX4 is a direct target gene of ARF5, but these are all indirect. When this statement is made, I do feel the authors should back this up with more direct evidence by e.g. ChIP Q-PCR or Y1H; as this is the most exciting and important part of the findings in my opinion.

The final statement in the discussion section, and hence also the main message of this manuscript, does not do justice to the manuscript, as other auxin signaling pathways have been shown to both regulate cell divisions and differentiation (e.g. Katayama et al Curr Biol. 2015;25(23):3144-50). The authors should consider a different statement at the end, or highlighting how their specific pathway is novel.

As a smaller general comment, I believe that in several instances, not all relevant references are cited. Moreover, adding a schematic as figure (with indication of the tissue specificity of all factors discussed) would improve the readability to a non-expert audience.

In figure 5, I can clearly see the boundaries leading up to the decision of the width of the phloem and cambium zone (orange and red line), but it is not clear at all where the xylem stops (and thus how long the blue line should be). This seems a bit arbitrary. How was this determined?

Reviewer #2 (Remarks to the Author):

While there maybe an interesting story in this manuscript I have some reservations about the rigour of the analysis to really determine what is happening and what the authors are actually looking at.

Firstly the scoring methods does not seem anywhere need adequate. Firstly there is a technical issue. Why are the authors looking a width. At the very least it would be trivial to analyse the image and then get some average once all of the regions of interest in the stem have been determined. Secondly why look at width at all, it would be trivial (and easier possibly) to measure cell number. This would be direct measure of cambial activity and/or phloem/xylem differentiation and should not be difficult to do.

It is unclear how the width was measured. I appreciate that the authors may have higher power and higher resolution images than those shown, but that fact that they have chosen to show these images means they are presumably above average or representative. Figure 5G is one of the clearest and the width of the phloem and cambium is clear, but how the transition from primary to secondary xylem seem entirely arbitrary. It is much less clear in other pictures. Similarly in figure 4H it is very hard to see any transition between primary and secondary xylem. Figure 3 is the most extreme there does not appear to be any consistent criteria applied.

The formation of a cambium at he base of stems late in development is presumably dependent upon both the activity of the cambium and response to external cues. In some pictures 3F it looks like there is a decrease in primary development. It looks like some lines exhibit very little secondary cell wall deposition in the primary intefascicular fibres. This may not be consistent and I accept it is hard to judge, but it would be useful to include some control to show that there are no other obvious defects in stem development. For example a section of the stem prior to secondary growth would at least suggest there is no decrease in primary xylem differentiation of interfascicular fibre formation.

The most fundamental question is whether the authors are looking at something that affects the timing of cambium initiation or cambium activity. There is excellent publicly available data from Poplar showing very high-resolution expression data. There is no appreciable variation in ARF3 or ARF4 expression across the phloem, cambium and xylem. The most obvious was to address cambial activity vs initiation would be to score the hypocotyl. As a means of scoring hypocotyl activity this has to be much easier there are far more vascular divisions in a mature hypocotyl so any difference in cambial activity are magnified and much easier to score.

Dear Reviewers

Many thanks for reconsidering your study "Spatial specificity of auxin responses coordinates wood formation" for publication in Nature Communications. We appreciated the positive attitude of the reviewers in response to our first submission and worked in the last months on addressing their valuable concerns and suggestions.

We especially believe that our added analyses of the short term dynamics of promoter reporters in response to auxin signalling modulation with tissue-specific resolution nicely confirm our initial hypothesis that ARF5 and auxin signalling in general acts locally on cambium-related genes. In particular, we could confirm that the *WOX4* promoter is locally repressed by both *bdl** and ARF5 induction. Importantly, after performing a computational binding site prediction, we were also able to demonstrate that the ARF5 protein binds directly a distinct motif in the *WOX4* promoter. Especially in light of these results, we believe that the quality of our study substantially increased and we are, thus, delighted to submit this revised version.

Reviewer #1

1) Using transcriptomics, the authors pinpoint WOX4 as a putative target to continue working on. It is however not entirely clear why this factor (which is well studied in the author's lab) is chosen and not one of the other known and interesting genes. Related to this, I found very little confirmation for this transcriptomics study. A second issue here is the fact that entire stem segments were taken. Although this is not a problem per se, it does become an issue when statements are made about the tissue specificity of factors. It is very plausible that a factor is induced in one tissue upon ectopic induction with GR, while it is repressed in another tissue. This would have implications in classifying genes into induced and repressed etc. In order to solve this (including the previous comment on confirmation), the authors should take a few marker lines and check that these are indeed induced or repressed in the respective tissues as is assumed at the moment. Many marker lines of genes in the lists are available in the community.

In response to this comment, we better justified the choice of *WOX4* in the text. Moreover, we included promoter reporter analyses confirming our transcriptome analyses. Exemplarily, we show that the *RUL1* promoter is, as predicted, locally activated by ARF5 Δ III/IV and repressed upon *bdl** induction. In comparison and again as predicted, the *PXY* promoter does not respond to those inductions but the *WOX4* promoter is repressed in both cases confirming our conclusion that *WOX4* responds differentially to distinct auxin signalling inputs.

2) As another comment related to this paragraph, it is not very clear to me at the moment why the authors propose ARF5 to function by promoting the transition of undifferentiated cells to differentiated xylem. Could the authors try to clarify this statement better with the evidence at hand, as it is one of the main conclusions in the manuscript. The similar statement is written down in the discussion section where it is stated that ARF5 promotes transition of stem cells to xylem cells based on the fact that genes such as ACL5, SACL3 and ATHB8 are recovered. In my opinion, an alternative hypothesis is that these genes are found simply because they are auxin dependent and this part of the

auxin signaling pathway goes through ARF5 (as has been shown in literature). Hence, I don't see how finding these genes contributes to the statement that is made.

We appreciate this comment and clarified this point in the revised version of the manuscript. We take the observation that known regulators of xylem differentiation are recovered in our cambium-related study, from which only some are direct ARF5 targets, and that specifically the cambium/xylem ratio is changed in *arf5* mutants, as indications that *ARF5* not only promotes xylem formation during primary organ development but also during secondary development. Because we believe that this finding substantially contributes to our understanding of cambium regulation, we put forward this conclusion in our manuscript.

3) As a final comment to this section, it would also help to better explain or hypothesize how and why pPXY:GR-ARF5deltaIII/IV and p35S:Myc-GR-bdl can have similar effects on gene expression upon induction with DEX (this could be related to the comment above about tissue specificity of the transcriptomics experiment when taking whole stem segments); and why ARF5 would act as transcriptional repressor as it is generally considered to be an activator ARF.

To give a possible scenario for why ARF5 acts as both, an activator and a repressor, we extended the discussion on that point. We especially want to mention that ARF5 has been described as a transcriptional repressor in other contexts as well (Zhao et al., 2010; Zhang et al., 2014) and that the classification of ARFs into activators and repressors is mostly based on their behaviour in transient expression systems but that their behaviour in their natural context is certainly more complex (Simonini et al., 2017).

4) In the paragraph about the link between WOX4 and ARF5, the authors performed all experiments possible to show that WOX4 is a direct target gene of ARF5, but these are all indirect. When this statement is made, I do feel the authors should back this up with more direct evidence by e.g. ChIP Q-PCR or Y1H; as this is the most exciting and important part of the findings in my opinion.

To address this undoubtedly central point, we followed two avenues in the last months in collaboration with the group of Jan Lohmann which has a long-standing history in performing studies on protein-DNA interactions and in analysing the function of ARF5. First, ChIP was performed taking advantage of plant lines expressing various tagged versions of ARF5 under its own promoter or the 35S promoter. However, as also other labs have experience in the past, ChIP for ARF5 was not successful and promoter binding could not even be confirmed for known direct ARF5 targets like ATHB8 and PIN1. As a second approach, we performed EMSA assays probing two predicted ARF binding sites in the WOX4 promoter. This showed that one of those motifs is bound by the ARF5 DNA binding domain, thereby confirming our assumptions.

5) The final statement in the discussion section, and hence also the main message of this manuscript, does not do justice to the manuscript, as other auxin signaling pathways have been shown to both regulate cell divisions and differentiation (e.g. Katayama et al Curr Biol. 2015;25(23):3144-50). The authors should consider a different statement at the end, or highlighting how their specific pathway is novel.

In response to this comment, we amended our statement by specifically mentioning the context of radial growth.

6) As a smaller general comment, I believe that in several instances, not all relevant references are cited. Moreover, adding a schematic as figure (with indication of the tissue specificity of all factors discussed) would improve the readability to a non-expert audience.

As a response, we now cite more relevant literature in the manuscript. We also added a scheme highlighting our concept of the role of ARF5 in cambium regulation and hope that this scheme clarifies our message.

7) In figure 5, I can clearly see the boundaries leading up to the decision of the width of the phloem and cambium zone (orange and red line), but it is not clear at all where the xylem stops (and thus how long the blue line should be). This seems a bit arbitrary. How was this determined?

To help the reader to follow the rational of our measurements, we added an explanation in Fig S3. See also point 8.

Reviewer #2 (Remarks to the Author):

8) Firstly the scoring methods does not seem anywhere need adequate. Firstly there is a technical issue. Why are the authors looking a width. At the very least it would be trivial to analyse the image and then get some average once all of the regions of interest in the stem have been determined. Secondly why look at width at all, it would be trivial (and easier possibly) to measure cell number. This would be direct measure of cambial activity and/or phloem/xylem differentiation and should not be difficult to do.

To address this concern, we added one figure explaining our measurement strategy (Fig. S3, see also point 7). In addition, we counted cells for one example (the mp mutant) to demonstrate that width and cell numbers correlate. Because this is a constant activity in our lab and the basis for many phenotypic characterizations, intensive discussions and investigations resulted in width measurement as the most reliable way to determine cambium activity. For example, automated image analyses and cell type detection failed so far due to the variation of the cellular outline and lack of contrast between tissues in sections (see below). Moreover, cell numbers especially in the cambium and phloem area is often not possible to determine reliably in sections due to the softer nature and the small size of those cells (see for example Fig 1J or 3K). Because we have also shown in the past (Gursansky et al., 2016) that width correlates strongly with cell numbers, we propose to generally stick to the way of measurements used in this study.

Three examples of segmentation and automated tissue recognition using the ilastik software. After processing an extensive training set, reliable automated tissue recognition is still non-reliable.

9) It is unclear how the width was measured. I appreciate that the authors may have higher power and higher resolution images than those shown, but that fact that they have chosen to show these

images means they are presumably above average or representative. Figure 5G is one of the clearest and the width of the phloem and cambium is clear, but how the transition from primary to secondary xylem seem entirely arbitrary. It is much less clear in other pictures. Similarly in figure 4H it is very hard to see any transition between primary and secondary xylem. Figure 3 is the most extreme there does not appear to be any consistent criteria applied.

See point 9.

10) The formation of a cambium at the base of stems late in development is presumably dependent upon both the activity of the cambium and response to external cues. In some pictures 3F it looks like there is a decrease in primary development. It looks like some lines exhibit very little secondary cell wall deposition in the primary interfascicular fibres. This may not be consistent and I accept it is hard to judge, but it would be useful to include some control to show that there are no other obvious defects in stem development. For example a section of the stem prior to secondary growth would at least suggest there is no decrease in primary xylem differentiation of interfascicular fibre formation.

As a response, we analysed primary stems of each mutant presented in this study (Fig. S7F-Q). We did not detect any major difference to the primary anatomy of wild type plants.

11) The most fundamental question is whether the authors are looking at something that affects the timing of cambium initiation or cambium activity. There is excellent publicly available data from Poplar showing very high-resolution expression data. There is no appreciable variation in ARF3 or ARF4 expression across the phloem, cambium and xylem. The most obvious was to address cambial activity vs initiation would be to score the hypocotyl. As a means of scoring hypocotyl activity this has to be much easier there are far more vascular divisions in a mature hypocotyl so any difference in cambial activity are magnified and much easier to score.

In response to this comment, we added analyses of mp mutant hypocotyls (Fig. S7R-T) which confirmed our conclusions. We also added pictures depicting the growth habitus of rooted strong mp mutants to demonstrate that those mutants develop stems similarly to wild type plants (Fig. S7D, E).

Reviewers' comments:

Reviewer #1 (Remarks to the Author):

Both reviewers have raised concerns about the way cambium activity has been measured. In this case, the authors perform these measurements by defining the width of phloem, cambium and xylem tissues. They have explained why they prefer their way to e.g. actually counting cell number etc. Although I have to admit that it is sometimes very difficult to see the boundaries between these tissue types, given their explanation, I have complete faith in the expertise of the authors that these measurements have been performed correctly. Having said this, I would encourage the authors to keep seeking for new dyes and protocols to make the distinction between the different tissue types even more clear for the non-experienced reader in the future.

Besides this and the minor textual comments below, the authors have addressed all my main concerns by including additional data and textual changes. I very much appreciate the additional efforts put in this manuscript to address my concerns.

- line 521: typo
- Cycloheximide is typically abbreviated as CHX, not Cyclo

Reviewer #2 (Remarks to the Author):

There are still some serious reservation about the scoring that are not addressed. One question is cell numbers vs width. Width is proxy for cambium activity and surely cell number is a better measure than width. Width appears to be measured at a single point, I cannot see the problem with manually counting cells even if it was only done for a single interfascicular region, it should be possible to get an average for single if reigns that would be better than a single width measurement. Secondly the point was also raised by the other reviewer. How are the limited of the widths measured? Even in the figures presented it is far from clear how where the limits are set. As far as I can tell there has been no attempt to address this directly and I am still unclear what object criteria were applied.

In the hypocotyl analysis I am not sure how showing mp mutants are small really says to support their central hypothesis. The organisation and initiation of phase II all look normal.

Dear reviewers,

we were happy to see that we were able to satisfy all the concerns of the first reviewer in response to our first submission. We are also confident that we are addressing sufficiently the concerns of the second reviewer in the second revised version of our manuscript. This is especially because we added more comparisons of domain width/cell numbers and show that there is perfect correlation (our new Fig. 7). We also base comparisons of domain relations now on cell number and not on extension anymore (Fig. 6H; Fig. 7C,D) following the reviewer's suggestion. Changed parts are labeled in yellow in the text. Find below a point-by-point response to all the concerns still raised.

Sincerely

Thomas

Reviewer #1 (Remarks to the Author):

Both reviewers have raised concerns about the way cambium activity has been measured. In this case, the authors perform these measurements by defining the width of phloem, cambium and xylem tissues. They have explained why they prefer their way to e.g. actually counting cell number etc. Although I have to admit that it is sometimes very difficult to see the boundaries between these tissue types, given their explanation, I have complete faith in the expertise of the authors that these measurements have been performed correctly. Having said this, I would encourage the authors to keep seeking for new dyes and protocols to make the distinction between the different tissue types even more clear for the non-experienced reader in the future.

1) Many thanks for this comment. We are aware of the need for a technical improvement of our phenotyping strategy and, although not being completed, we are currently working intensively on that topic.

Besides this and the minor textual comments below, the authors have addressed all my main concerns by including additional data and textual changes. I very much appreciate the additional efforts put in this manuscript to address my concerns.

- line 521: typo

- Cycloheximide is typically abbreviated as CHX, not Cyclo

2) We implemented the two minor suggestions in the current version of the manuscript (highlighted in yellow).

Reviewer #2 (Remarks to the Author):

There are still some serious reservation about the scoring that are not addressed. One question is cell numbers vs width. Width is proxy for cambium activity and surely cell number is a better measure than width. Width appears to be measured at a single point, I cannot see the problem with manually counting cells even if it was only done for a single interfascicular region, it should be possible to get an average for single if reigns that would be better than a single width measurement. Secondly the point was also raised by the other reviewer. How are the limited of the widths measured? Even in the

figures presented it is far from clear how where the limits are set. As far as I can tell there has been no attempt to address this directly and I am still unclear what object criteria were applied.

- 3) We again appreciate the critical evaluation of our phenotyping strategy. As a direct response we added more cell number countings of *pPXY:GR-ARF5 Δ III/IV* lines (now in Figure 7) which similarly support our conclusions that ARF5 mostly acts on the transition from cambium to xylem cells. We also base comparisons of domain relations now on cell number and not on extension anymore (Fig. 6H; Fig. 7C,D) addressing exactly the reviewer's concern. We also want to politely point out, that we indeed added a in-depth description of our measurement strategy (Fig. S3) in response to this concern in the previous version of the manuscript. In this description we explain in detail how the distal and proximal margin of the zone measured is determined. We also added in the last version cell number calculations for the *arf5* mutant to address the same concern. In addition, we showed previously (Gursansky et al., 2016) and in this study that width correlates perfectly with the number of cells in interfascicular regions. Moreover, also in other studies we applied successfully the same strategy for identifying biologically relevant phenotypic differences (Sehr et al., 2010; Agusti et al., 2011a, b; Suer et al., 2011). Therefore, we are highly confident that width, in combination with cell number calculations as presented in this study, is a rigorous way of determining cambium activity. We also want to emphasize that we always measure (width) and count (cell numbers) in several interfascicular regions and determine the average also applying proper statistical methods as explained in the M+M section.

In the hypocotyl analysis I am not sure how showing *mp* mutants are small really says to support their central hypothesis. The organisation and initiation of phase II all look normal.

- 4) We want to point out that the *mp* (*arf5*) mutant is, in fact, not smaller but larger in size in the hypocotyl which confirms our analyses performed in the inflorescence shoot. We believe that distinguishing between phase I and phase II of xylem development goes considerably beyond the scope of this study.